# Ranking Policy Gradient

**Kaixiang Lin**
Department of Computer Science and Engineering
Michigan State University
East Lansing, MI 48824-4403, USA
`linkaixi@msu.edu`

**Jiayu Zhou**
Department of Computer Science and Engineering
Michigan State University
East Lansing, MI 48824-4403, USA
`jiayuz@msu.edu`

## Abstract

Sample inefficiency is a long-lasting problem in reinforcement learning (RL). The state-of-the-art estimates the optimal action values while it usually involves an extensive search over the state-action space and unstable optimization. Towards the sample-efficient RL, we propose *ranking policy gradient* (RPG), a policy gradient method that learns the optimal rank of a set of discrete actions. To accelerate the learning of policy gradient methods, we establish the equivalence between maximizing the lower bound of return and imitating a near-optimal policy without accessing any oracles. These results lead to a general off-policy learning framework, which preserves the optimality, reduces variance, and improves the sample-efficiency. We conduct extensive experiments showing that when consolidating with the off-policy learning framework, RPG substantially reduces the sample complexity, comparing to the state-of-the-art.

## 1 Introduction

One of the major challenges in reinforcement learning (RL) is the high *sample complexity* (Kakade et al., 2003), which is the number of samples must be collected to conduct successful learning. There are different reasons leading to poor sample efficiency of RL (Yu, 2018). Because policy gradient algorithms directly optimizing return estimated from rollouts (e.g., Reinforce (Williams, 1992)) could suffer from high variance (Sutton & Barto, 2018), value function baselines were introduced by actor-critic methods to reduce the variance and improve the sample-efficiency. However, since a value function is associated with a certain policy, the samples collected by former policies cannot be readily used without complicated manipulations (Degris et al., 2012) and extensive parameter tuning (Nachum et al., 2017). Such an on-policy requirement increases the difficulty of sample-efficient learning.

On the other hand, off-policy methods, such as one-step $Q$-learning (Watkins & Dayan, 1992) and variants of deep $Q$ networks (DQN) (Mnih et al., 2015; Hessel et al., 2017; Dabney et al., 2018; Van Hasselt et al., 2016; Schaul et al., 2015), enjoys the advantage of learning from any trajectory sampled from the same environment (i.e., off-policy learning), are currently among the most sample-efficient algorithms. These algorithms, however, often require extensive searching (Bertsekas & Tsitsiklis, 1996, Chap. 5) over the large state-action space to estimate the optimal action value function. Another deficiency is that, the combination of off-policy learning, bootstrapping, and function approximation, making up what Sutton & Barto (2018) called the "deadly triad", can easily lead to unstable or even divergent learning (Sutton & Barto, 2018, Chap. 11). These inherent issues limit their sample-efficiency.

Towards addressing the aforementioned challenge, we approach the sample-efficient reinforcement learning from a ranking perspective. Instead of estimating optimal action value function, we concentrate on learning optimal rank of actions. The rank of actions depends on the *relative action values*. As long as the relative action values preserve the same rank of actions as the optimal action values ($Q$-values), we choose the same optimal action. To learn optimal relative action values, we propose the *ranking policy gradient (RPG)* that optimizes the actions' rank with respect to the long-term reward by learning the pairwise relationship among actions.

Ranking Policy Gradient (RPG) that directly optimizes relative action values to maximize the return is a policy gradient method. The track of off-policy actor-critic methods (Degris et al., 2012; Gu et al., 2016; Wang et al., 2016) have made substantial progress on improving the sample-efficiency

of policy gradient. However, the fundamental difficulty of learning stability associated with the bias-variance trade-off remains (Nachum et al., 2017). In this work, we first exploit the equivalence between RL optimizing the lower bound of return and supervised learning that imitates a specific optimal policy. Build upon this theoretical foundation, we propose a general off-policy learning framework that equips the generalized policy iteration (Sutton & Barto, 2018, Chap. 4) with an external step of supervised learning. The proposed off-policy learning not only enjoys the property of optimality preserving (unbiasedness), but also largely reduces the variance of policy gradient because of its independence of the horizon and reward scale. Besides, we empirically show that there is a trade-off between optimality and sample-efficiency. Last but not least, we demonstrate that the proposed approach, consolidating the RPG with off-policy learning, significantly outperforms the state-of-the-art (Hessel et al., 2017; Bellemare et al., 2017; Dabney et al., 2018; Mnih et al., 2015).

## 2 RELATED WORKS

*Sample Efficiency.* The sample efficient reinforcement learning can be roughly divided into two categories. The first category includes variants of $Q$-learning (Mnih et al., 2015; Schaul et al., 2015; Van Hasselt et al., 2016; Hessel et al., 2017). The main advantage of $Q$-learning methods is the use of off-policy learning, which is essential towards sample efficiency. The representative DQN (Mnih et al., 2015) introduced deep neural network in $Q$-learning, which further inspried a track of successful DQN variants such as Double DQN (Van Hasselt et al., 2016), Dueling networks (Wang et al., 2015), prioritized experience replay (Schaul et al., 2015), and RAINBOW (Hessel et al., 2017). The second category is the actor-critic approaches. Most of recent works (Degris et al., 2012; Wang et al., 2016; Gruslys et al., 2018) in this category leverage importance sampling by re-weighting the samples to correct the estimation bias and reduce variance. Its main advantage is in the wall-clock times due to the distributed framework, firstly presented in (Mnih et al., 2016), instead of the sample-efficiency. As of the time of writing, the variants of DQN (Hessel et al., 2017; Dabney et al., 2018; Bellemare et al., 2017; Schaul et al., 2015; Van Hasselt et al., 2016) are among the algorithms of most sample efficiency, which are adopted as our baselines for comparison.

*RL as supervised learning.* Numerous amount of works have developed the connections between RL and supervised learning such as Expectation-Maximization algorithms (Dayan & Hinton, 1997; Peters & Schaal, 2007; Kober & Peters, 2009; Abdolmaleki et al., 2018), Entropy-Regularized RL (Oh et al., 2018; Haarnoja et al., 2018), and Interactive Imitation Learning (IIL) (Daumé et al., 2009; Syed & Schapire, 2010; Ross & Bagnell, 2010; Ross et al., 2011; Sun et al., 2017; Hester et al., 2018; Osa et al., 2018). EM-based approaches utilize the probabilistic framework to transfer RL maximizing lower bound of return as a re-weighted regression problem while it requires on-policy estimation on the expectation step. Entropy-Regularized RL optimizing entropy augmented objectives can lead to off-policy learning without the usage of importance sampling while it converges to soft optimality (Haarnoja et al., 2018).

Of the three tracks in prior works, the IIL is most closely related to our work. The IIL works firstly pointed out the connection between imitation learning and reinforcement learning (Ross & Bagnell, 2010; Syed & Schapire, 2010; Ross et al., 2011) and explore the idea of facilitating reinforcement learning by imitating experts. However, most of imitation learning algorithms assume the access to the expert policy or demonstrations. Our off-policy learning framework can be interpreted as an online imitation learning approach that constructs expert demonstrations during the exploration without soliciting experts, and conducts supervised learning to maximize return at the same time.

In conclusion, our approach is different from the prior work in terms of at least one of the following aspects: objectives, oracle assumptions, the optimality of learned policy, and on-policy requirement. More concretely, the proposed method is able to learn both deterministic and stochastic optimal policy in terms of long-term reward, without access to the oracle (such as expert policy or expert demonstration) and it can be trained both empirically and theoretically in an off-policy fashion. Due to the space limits, we defer the detailed discussion of the related work in the Appendix Section 10.1.

## 3 NOTATIONS AND PROBLEM SETTING

In this paper, we consider a finite horizon $T$, discrete time Markov Decision Process (MDP) with a finite discrete state space $\mathcal{S}$ and for each state $s \in \mathcal{S}$, the action space $\mathcal{A}_s$ is finite. The environment dynamics is denoted as $\mathbf{P} = \{p(s'|s,a), \forall s, s' \in \mathcal{S}, a \in \mathcal{A}_s\}$. We note that the dimension of action space can vary given different states. We use $m = \max_s \|\mathcal{A}_s\|$ to denote the maximal action dimension among all possible states. Our goal is to maximize the expected sum of rewards, or return

$J(\theta) = \mathbf{E}_{\tau,\pi_\theta}[\sum_{t=1}^{T} r(s_t, a_t)]$, where $|r(s,a)| < \infty, \forall s, a$. In this case, the optimal deterministic Markovian policy always exists (Puterman, 2014)[Proposition 4.4.3]. The upper bound of trajectory reward ($r(\tau)$) is denoted as $R_{\max} = \max_\tau r(\tau)$. A comprehensive list of notations are elaborated in the Appendix Table 1.

## 4 RANKING POLICY GRADIENT

Value function estimation is widely used in advanced RL algorithms (Mnih et al., 2015; 2016; Schulman et al., 2017; Gruslys et al., 2018; Hessel et al., 2017; Dabney et al., 2018) to facilitate the learning process. In practice, the on-policy requirement of value function estimations in actor-critic methods has largely increased the difficulty of sample-efficient learning (Degris et al., 2012; Gruslys et al., 2018). With the advantage of off-policy learning, the DQN (Mnih et al., 2015) variants are currently among the most sample-efficient algorithms (Hessel et al., 2017; Dabney et al., 2018; Bellemare et al., 2017). For complicated tasks, the value function can align with the relative relationship of action's return, but the absolute values are hardly accurate (Mnih et al., 2015; Ilyas et al., 2018).

The above observations motivate us to look at the decision phase of RL from a different prospect: Given a state, the decision making is to perform a *relative comparison* over available actions and then choose the best action, which can lead to relatively higher return than others. Therefore, an alternative solution is to learn the optimal rank of the actions. In this section, we show how to optimize the rank of actions to maximize the return, and thus avoid the necessity of accurate estimation for optimal action value function. To learn the rank of actions, we focus on learning *relative action value* ($\lambda$-values), defined as follows:

**Definition 1** (Relative action value ($\lambda$-values))**.** *For a state $s$, the relative action values of $m$ actions ($\lambda(s, a_k), k = 1, ..., m$) is a list of scores that denotes the rank of actions. If $\lambda(s, a_i) > \lambda(s, a_j)$, then action $a_i$ is ranked higher than action $a_j$.*

The optimal relative action values should preserve the same optimal action as the optimal action values:

$$\arg\max_a \lambda(s, a) = \arg\max_a Q^{\pi_*}(s, a)$$

where $Q^{\pi_*}(s, a_i)$ and $\lambda(s, a_i)$ represent the optimal action value and the relative action value of action $a_i$, respectively. We omit the model parameter $\theta$ in $\lambda_\theta(s, a_i)$ for concise presentation.

**Remark 1.** *The $\lambda$-values are different from the advantage function $A^\pi(s, a) = Q^\pi(s, a) - V^\pi(s)$. The advantage functions quantitatively show the difference of return taking different actions following the current policy $\pi$. The $\lambda$-values only determine the relative order of actions and its magnitudes are not the estimations of returns.*

To learn the $\lambda$-values, we can construct a probabilistic model of $\lambda$-values such that the best action has the highest probability to be selected than others. Inspired by learning to rank (Burges et al., 2005), we consider the pairwise relationship among all actions, by modeling the probability (denoted as $p_{ij}$) of an action $a_i$ to be ranked higher than any action $a_j$ as follows:

$$p_{ij} = \frac{\exp(\lambda(s, a_i) - \lambda(s, a_j))}{1 + \exp(\lambda(s, a_i) - \lambda(s, a_j))}, \tag{1}$$

where $p_{ij} = 0.5$ means the relative action value of $a_i$ is same as that of the action $a_j$, $p_{ij} > 0.5$ indicates that the action $a_i$ is ranked higher than $a_j$. Given the independent Assumption 1, we can represent the probability of selecting one action as the multiplication of a set of pairwise probabilities in Eq (1). Formally, we define the pairwise ranking policy in Eq (2). Please refer to Section 10.10 in the Appendix for the discussions on feasibility of Assumption 1.

**Definition 2.** *The pairwise ranking policy is defined as:*

$$\pi(a = a_i|s) = \Pi_{j=1,j\neq i}^{m} p_{ij}, \tag{2}$$

*where the $p_{ij}$ is defined in Eq (1). The probability depends on the relative action values $q = [\lambda_1, ..., \lambda_m]$. The highest relative action value leads to the highest probability to be selected.*

**Assumption 1.** *For a state $s$, the set of events $E = \{e_{ij}|\forall i \neq j\}$ are conditionally independent, where $e_{ij}$ denotes the event that action $a_i$ is ranked higher than action $a_j$. The independence of the events is conditioned on a MDP and a stationary policy.*

Our ultimate goal is to maximize the long-term reward through optimizing the pairwise ranking policy or equivalently optimizing pairwise relationship among the action pairs. Ideally, we would like the pairwise ranking policy selects the best action with the highest probability and the highest $\lambda$-value. To achieve this goal, we resort to the policy gradient method. Formally, we propose the ranking policy gradient method (RPG), as shown in Theorem 1.

**Theorem 1** (Ranking Policy Gradient Theorem). *For any MDP, the gradient of the expected long-term reward $J(\theta) = \sum_\tau p_\theta(\tau)r(\tau)$ w.r.t. the parameter $\theta$ of a pairwise ranking policy (Def 2) can be approximated by:*

$$\nabla_\theta J(\theta) \approx \mathbf{E}_{\tau \sim \pi_\theta}\left[\sum_{t=1}^{T} \nabla_\theta \left(\sum_{j=1, j\neq i}^{m} (\lambda_i - \lambda_j)/2\right) r(\tau)\right], \qquad (3)$$

*and the deterministic pairwise ranking policy $\pi_\theta$ is: $a = \arg\max_i \lambda_i, \ i = 1, \ldots, m$, where $\lambda_i$ denotes the relative action value of action $a_i$ ($\lambda_\theta(s_t, a_t)$, $a_i = a_t$), $s_t$ and $a_t$ denotes the $t$-th state-action pair in trajectory $\tau$, $\lambda_j, \forall j \neq i$ denote the relative action values of all other actions that were not taken given state $s_t$ in trajectory $\tau$, i.e., $\lambda_\theta(s_t, a_j)$, $\forall a_j \neq a_t$.*

The proof of Theorem 1 is available in Appendix Section 10.2. Theorem 1 states that optimizing the discrepancy between the relative action values of the best action and all other actions, is optimizing the pairwise relationships that maximize the return. One limitation of RPG is that it is not convenient for the tasks where only optimal stochastic policies exist since the pairwise ranking policy takes extra efforts to construct a probability distribution [see Section 10.3 in Appendix]. In order to learn the stochastic policy, we introduce Listwise Policy Gradient (LPG) that optimizes the probability of ranking a specific action on the top of a set of actions, with respect to the return. In the context of RL, this top one probability is the probability of action $a_i$ to be chosen, which is equal to the sum of probability all possible permutations that map action $a_i$ in the top. Inspired by listwise learning to rank approach (Cao et al., 2007), the top one probability can be modeled by the softmax function. Therefore, LPG is equivalent to the REINFORCE (Williams, 1992) algorithm with a softmax layer. LPG provides another interpretation of REINFORCE algorithm from the perspective of learning the optimal ranking and enables the learning of both deterministic policy and stochastic policy. Due to the space limit, we defer the detailed description of LPG in Appendix Section 10.4.

To this end, seeking sample-efficiency motivates us to learn the relative relationship (RPG (Theorem 1) and LPG (Theorem 4)) of actions, instead of seeking accurate estimation of optimal action values and then choosing action greedily. However, both of the RPG and LPG belong to policy gradient methods, which suffers from large variance and the on-policy learning requirement (Sutton & Barto, 2018). Therefore, the direct implementation of RPG or LPG is still far from sample-efficient. In the next section, we will describe a general off-policy learning framework empowered by supervised learning, which provides an alternative way to accelerate learning, preserve optimality, and reduce variance.

## 5 OFF-POLICY LEARNING AS SUPERVISED LEARNING

In this section, we discuss the connections and discrepancies between RL and supervised learning, and our results lead to a sample-efficient off-policy learning paradigm for RL. The main result in this section is Theorem 2, which casts the problem of maximizing the lower bound of return into a supervised learning problem, given one relatively mild Assumption 2 and practical Assumptions 1,3. As we show by Lemma 4 in the Appendix that assumptions are valid in a range of RL tasks. The central idea is to collect only the near-optimal trajectories when the learning agent interacts with the environment, and imitate the near-optimal policy by maximizing the log likelihood of the state-action pairs from near-optimal trajectories. With the road map in mind, we then begin to introduce our approach as follows.

In a discrete action MDP with finite states and horizon, given the near-optimal policy $\pi_*$, the stationary state distribution is given by: $p_{\pi_*}(s) = \sum_\tau p(s|\tau)p_{\pi_*}(\tau)$, where $p(s|\tau)$ is the probability of a certain state given a specific trajectory $\tau$ and is not associated with any policies, and only

$p_{\pi_*}(\tau)$ is related to the policy parameters. The stationary distribution of state-action pairs is thus: $p_{\pi_*}(s,a) = p_{\pi_*}(s)\pi_*(a|s)$. In this section, we consider the MDP that each initial state will lead to at least one (near)-optimal trajectory. For a more general case, please refer to the discussion in Appendix 10.5. In order to connect supervised learning (i.e., imitating a near-optimal policy) with RL and enable sample-efficient off-policy learning, we first introduce the trajectory reward shaping (TRS), defined as follows:

**Definition 3** (Trajectory Reward Shaping, TRS). *Given a fixed trajectory $\tau$, its trajectory reward is shaped as follows:*

$$w(\tau) = \left\{ \begin{array}{ll} 1, & \text{if } r(\tau) \geq c \\ 0, & o.w. \end{array} \right.$$

*where $c = R_{\max} - \epsilon$ is a problem-dependent near-optimal trajectory reward threshold that indicates the least reward of near-optimal trajectory, $\epsilon \geq 0$ and $\epsilon \ll R_{\max}$. We denote the set of all possible near-optimal trajectories as $\mathcal{T} = \{\tau | w(\tau) = 1\}$, i.e., $w(\tau) = 1, \forall \tau \in \mathcal{T}$.*

**Remark 2.** *The threshold $c$ indicates a trade-off between the sample-efficiency and the optimality. The higher the threshold, the less frequently it will hit the near-optimal trajectories during exploration, which means it has higher sample complexity, while the final performance is better (see Figure 3).*

**Remark 3.** *The trajectory reward can be reshaped to any positive functions that are not related to policy parameter $\theta$. For example, if we set $w(\tau) = r(\tau)$, the conclusions in this section still hold (see Eq (38) in Appendix, Section 10.6). For the sake of simplicity, we set $w(\tau) = 1$.*

Different from the reward shaping works (Ng et al., 1999), we directly shape the trajectory reward, which will enable the smooth transform from RL to SL. After shaping the trajectory reward, we can transfer the goal of RL from maximizing the return to maximize the long-term performance (Def 4).

**Definition 4** (Long-term Performance).

$$\sum_{\tau} p_{\theta}(\tau)w(\tau) \tag{4}$$

*The long-term performance is the expected shaped trajectory reward, as shown in Eq (4). By Def 3, the expectation over all trajectories is the equal to that over the near-optimal trajectories in $\mathcal{T}$, i.e., $\sum_{\tau} p_{\theta}(\tau)w(\tau) = \sum_{\tau \in \mathcal{T}} p_{\theta}(\tau)w(\tau)$.*

The optimality is preserved after trajectory reward shaping ($\epsilon = 0, c = R_{\max}$) since the optimal policy $\pi_*$ maximizing long-term performance is also an optimal policy for original MDP, i.e., $\sum_{\tau} p_{\pi_*}(\tau)r(\tau) = \sum_{\tau \in \mathcal{T}} p_{\pi_*}(\tau)r(\tau) = R_{\max}$, where $\pi_* = \arg\max_{\pi_\theta} \sum_{\tau} p_{\pi_\theta}(\tau)w(\tau)$ and $p_{\pi_*}(\tau) = 0, \forall \tau \notin \mathcal{T}$ (see Lemma 2 in Appendix 10.6). Similarly, when $\epsilon > 0$, the optimal policy after trajectory reward shaping is a near-optimal policy for original MDP. Note that most policy gradient methods use softmax function, in which we have $\exists \tau \notin \mathcal{T}, p_{\pi_\theta}(\tau) > 0$ (see Lemma 3 in Appendix 10.6). Therefore when softmax is used to model a policy, it will not converge to an exact optimal policy. On the other hand, ideally, the discrepancy of the performance between them can be arbitrarily small based on the universal approximation (Hornik et al., 1989) with general conditions on the activation function and Theorem 1. in (Syed & Schapire, 2010).

Essentially, we use TRS to filter out near-optimal trajectories and then we maximize the probabilities of near-optimal trajectories to maximize the long-term performance. This procedure can be approximated by maximizing the log-likelihood of near-optimal state-action pairs, which is a supervised learning problem. Before we state our main results, we first introduce the definition of uniformly near-optimal policy (Def 5) and a prerequisite (Asm. 2) specifying the applicability of the results.

**Definition 5** (Uniformly Near-Optimal Policy, UNOP). *The Uniformly Near-Optimal Policy $\pi_*$ is the policy whose probability distribution over near-optimal trajectories ($\mathcal{T}$) is a uniform distribution. i.e. $p_{\pi_*}(\tau) = \frac{1}{|\mathcal{T}|}, \forall \tau \in \mathcal{T}$, where $|\mathcal{T}|$ is the number of near-optimal trajectories. When we set $c = R_{\max}$, it is an optimal policy in terms of both maximizing return and long-term performance. In the case of $c = R_{\max}$, the corresponding uniform policy is an optimal policy, we denote this type of optimal policy as uniformly optimal policy (UOP).*

**Assumption 2** (Existence of Uniformly Near-Optimal Policy). *We assume the existence of Uniformly Near-Optimal Policy (Def 5).*

Based on Lemma 4 in Appendix Section 10.9, Assumption 2 is satisfied for certain MDPs that have deterministic dynamics. Other than Assumption 2, all other assumptions in this work (Assumptions 1,3) can almost always be satisfied in practice, based on empirical observation. With these relatively mild assumptions, we present the following long-term performance theorem, which shows the close connection between supervised learning and RL.

**Theorem 2** (Long-term Performance Theorem). *Maximizing the lower bound of expected long-term performance (Eq (4)) is maximizing the log-likelihood of state-action pairs sampled from an uniformly (near)-optimal policy $\pi_*$, which is a supervised learning problem:*

$$\arg\max_\theta \sum_{s,a} p_{\pi_*}(s,a) \log \pi_\theta(a|s) \tag{5}$$

*The optimal policy of maximizing the lower bound is also the optimal policy of maximizing the long-term performance and the return.*

**Remark 4.** *It is worth noting that Theorem 2 does not require a uniformly near-optimal policy $\pi_*$ to be deterministic. The only requirement is the existence of a uniformly near-optimal policy.*

**Remark 5.** *Maximizing the lower bound of long-term performance is to maximize the lower bound of long-term reward since we can set $w(\tau) = r(\tau)$ and $\sum_\tau p_\theta(\tau) r(\tau) \geq \sum_\tau p_\theta(\tau) w(\tau)$. An optimal policy of maximizing this lower bound is also an optimal policy of maximizing the long-term performance when $c = R_{\max}$, thus maximizing the return.*

The proof of Theorem 2 can be found in Appendix, Section 10.6. Theorem 2 indicates that we break the dependency between current policy $\pi_\theta$ and the environment dynamics, which means off-policy learning is able to be conducted by the above supervised learning approach. Furthermore, we point out that there is a potential discrepancy between imitating UNOP by maximizing log likelihood (even when the optimal policy's samples are given) and the reinforcement learning since we are maximizing a lower bound of expected long-term performance (or equivalently the return over the near-optimal trajectories only) instead of return over all trajectories. In practice, the state-action pairs from an optimal policy is hard to construct while the uniform characteristic of UNOP can alleviate this issue (see Sec 6). Towards sample-efficient RL, we apply Theorem 2 to RPG, which reduces the ranking policy gradient to a classification problem by Corollary 1.

**Corollary 1** (Ranking performance policy gradient). *Optimizing the lower bound of expected long-term performance (defined in Eq (4)) using pairwise ranking policy (Eq (2)) can be approximately optimized by the following loss:*

$$\min_\theta \sum_{s,a_i} p_{\pi_*}(s,a_i) \left( \sum_{j=1,j\neq i}^m \max(0, margin + \lambda(s,a_j) - \lambda(s,a_i)) \right), \tag{6}$$

*where margin is a small positive value. We set margin equal to one in our experiments.*

The proof of Corollary 1 can be found in Appendix, section 10.7. Similarly, we can reduce LPG to a classification problem (see Appendix 10.7.1). One advantage of casting RL to SL is variance reduction. With the proposed off-policy supervised learning, we can reduce the upper bound of the policy gradient variance, as shown in the Corollary 2. Before introducing the variance reduction results, we first make the following standard assumption similar to (Degris et al., 2012, A1). Furthermore, the assumption is guaranteed for bounded continuously differentiable policy such as softmax function.

**Assumption 3.** *We assume the maximum norm of policy gradient is finite, i.e.*

$$\exists\, C < \infty,\ s.t.\ \|\nabla_\theta \log \pi_\theta(a|s)\|_\infty \leq C, \forall\, s \in \mathcal{S}, a \in \mathcal{A}_s$$

**Corollary 2** (Policy gradient variance reduction). *The upper bound of the variance of each dimension of policy gradient is $O(T^2 C^2 M^2)$. The upper bound of gradient variance of maximizing the lower bound of long-term performance Eq (5) is $O(C^2)$, where $C$ is the maximum norm of log gradient based on Assumption 3, $M$ is the maximum absolute value of trajectory reward (i.e., $M \geq |r(\tau)|, \forall \tau$), and $T$ is the horizon. The upper bound of gradient variance by supervised learning compared to that of the regular policy gradient is reduced by an order of $O(T^2 M^2)$, given $M > 1, T > 1$, which is a very common situation in practice, and a stationary policy.*

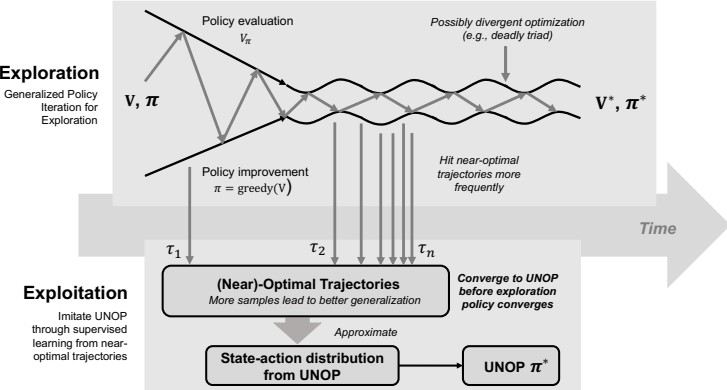

Figure 1: The off-policy learning as supervised learning framework for general policy gradient methods.

The proof of Corollary 2 can be found in Appendix 10.8. This corollary shows that the variance of regular policy gradient is upper-bounded by the square of time horizon and the maximum trajectory reward. It is aligned with our intuition and empirical observation: the longer the horizon the harder the learning. Also, the common reward shaping tricks such as truncating the reward to $[-1, 1]$ (Castro et al., 2018) can help the learning since it reduces variance by decreasing the range of trajectory reward. With supervised learning, we concentrate the difficulty of long-time horizon into the exploration phase, which is an inevitable issue for all RL algorithms, and we drop the dependence on $T$ and $M$ for policy variance. Thus, it is more stable and efficient to train the policy using supervised learning. One potential limitation of this method is that the trajectory reward threshold $c$ is task-specific, which is crucial to the final performance and sample-efficiency. In many applications such as Dialogue system (Li et al., 2017), recommender system (Melville & Sindhwani, 2011), etc., we design the reward function to guide the learning process, in which $c$ is naturally known. For the cases that we have no prior knowledge on the reward function of MDP, we treat $c$ as a tuning parameter to balance the optimality and efficiency, as we empirically verified in Figure 3. The major theoretical uncertainty on general tasks is the existence of a uniformly near-optimal policy, which is negligible to the empirical performance. The rigorous theoretical analysis of this problem is beyond the scope of this work.

## 6 An Algorithmic Framework for Off-Policy Learning

Based on the discussions in Section 5, we exploit the advantage of reducing RL into supervised learning via a proposed two-stages off-policy learning framework. As we illustrated in Figure 1, the proposed framework contains the following two stages:

**Generalized Policy Iteration for Exploration.** The goal of the exploration stage is to collect different near-optimal trajectories as frequently as possible. Under the off-policy framework, the exploration agent and the learning agent can be separated. Therefore, any existing RL algorithm can be used during the exploration. The principle of this framework is using the most advanced RL agents as an exploration strategy in order to collect more near-optimal trajectories and leave the policy learning to the supervision stage.

**Supervision.** In this stage, we imitate the uniformly near-optimal policy, UNOP (Def 5). Although we have no access to the UNOP, we can approximate the state-action distribution from UNOP by collecting the near-optimal trajectories only. The near-optimal samples are constructed online and we are not given any expert demonstration or expert policy beforehand. This step provides a sample-efficient approach to conduct exploitation, which enjoys the superiority of stability (Figure 2), variance reduction (Corollary 2), and optimality preserving (Theorem 2).

The two-stage algorithmic framework can be directly incorporated in RPG and LPG to improve sample efficiency. The implementation of RPG is given in Algorithm 1, and LPG follows the same procedure except for the difference in the loss function. The main requirement of Alg. 1 is on the exploration efficiency and the MDP structure. During the exploration stage, a sufficient amount of the different near-optimal trajectories need to be collected for constructing a representative supervised learning training dataset. Theoretically, this requirement always holds [see Appendix Section 10.9, Lemma 5], while the number of episodes explored could be prohibitively large, which makes this

algorithm sample-inefficient. This could be a practical concern of the proposed algorithm. However, according to our extensive empirical observations, we notice that long before the value function based state-of-the-art converges to near-optimal performance, enough amount of near-optimal trajectories are already explored.

Therefore, we point out that instead of estimating optimal action value functions and then choosing action greedily, using value function to facilitate the exploration and imitating UNOP is a more sample-efficient approach. As illustrated in Figure 1, value based methods with off-policy learning, bootstrapping, and function approximation could lead to a divergent optimization (Sutton & Barto, 2018, Chap. 11). In contrast to resolving the instability, we circumvent this issue via constructing a stationary target using the samples from (near)-optimal trajectories, and perform imitation learning. This two-stage approach can avoid the extensive exploration of the suboptimal state-action space and reduce the substantial number of samples needed for estimating optimal action values. In the MDP where we have a high probability of hitting the near-optimal trajectories (such as PONG), the supervision stage can further facilitate the exploration. It should be emphasized that our work focuses on improving the sample-efficiency through more effective exploitation, rather than developing novel exploration method. Please refer to the Appendix Section 10.11 for more discussion on exploration efficiency.

## 7 EXPERIMENTAL RESULTS

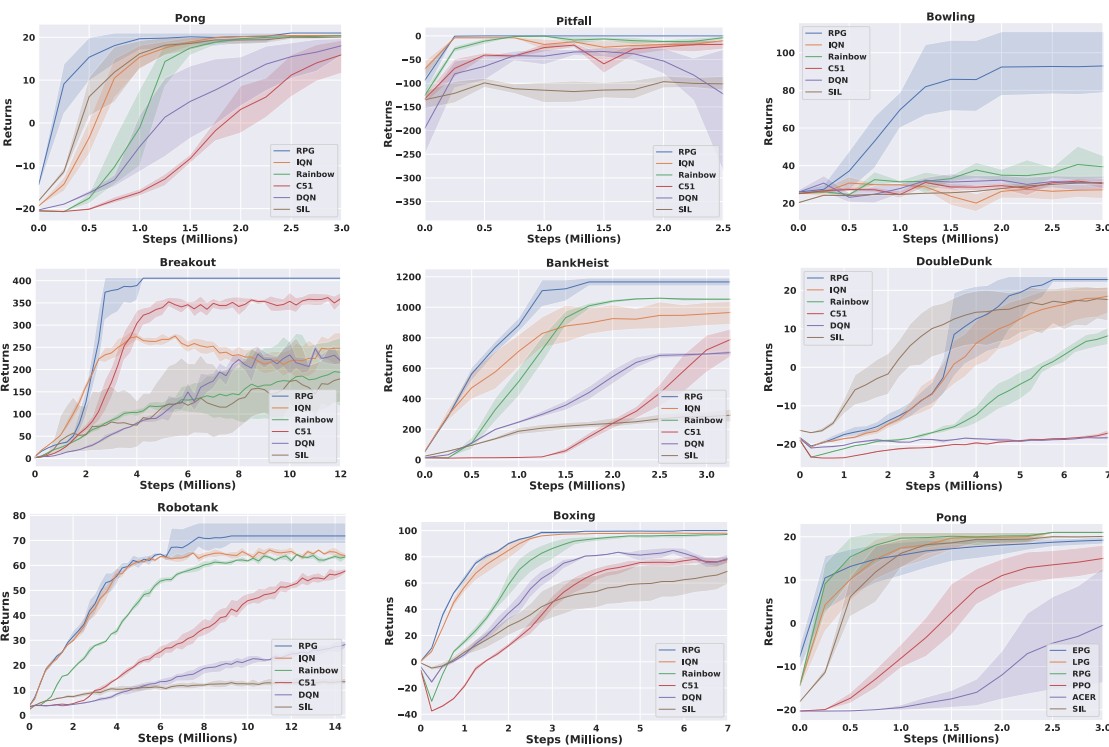

Figure 2: The training curves of the proposed RPG and state-of-the-art. All results are averaged over random seeds from 1 to 5. The $x$-axis represents the number of steps interacting with the environment (we update the model every four steps) and the $y$-axis represents the averaged training episodic return. The error bars are plotted with a confidence interval of 95%.

To evaluate the sample-efficiency of Ranking Policy Gradient (RPG), we focus on Atari 2600 games in OpenAI gym Bellemare et al. (2013); Brockman et al. (2016), without randomly repeating the previous action. We compare our method with the state-of-the-art baselines including DQN Mnih et al. (2015), C51 Bellemare et al. (2017), IQN Dabney et al. (2018), RAINBOW Hessel et al. (2017), and self-imitation learning (SIL) Oh et al. (2018). For reproducibility, we use the implementation provided in Dopamine framework[1] Castro et al. (2018) for all baselines and proposed methods, except

---

[1]https://github.com/google/dopamine

for SIL using the official implementation [2]. Following the standard evaluation protocol Oh et al. (2018); Hessel et al. (2017); Dabney et al. (2018); Bellemare et al. (2017), we report the training performance of all baselines as the increase of interactions with the environment, or proportionally the number of training iterations. We run the algorithms with five random seeds and report the average rewards with 95% confidence intervals. The implementation details of the proposed RPG and its variants are given as follows[3]:

**EPG**: EPG is the stochastic listwise policy gradient (see Appendix Eq (18)) incorporated with the proposed off-policy learning. More concretely, we apply trajectory reward shaping (TRS, Def 3) to all trajectories encountered during exploration and train vanilla policy gradient using the off-policy samples. This is equivalent to minimizing the cross-entropy loss (see Appendix Eq (69)) over the near-optimal trajectories.

**LPG**: LPG is the deterministic listwise policy gradient with the proposed off-policy learning. The only difference between EPG and LPG is that LPG chooses action deterministically (see Appendix Eq (17)) during evaluation.

**RPG**: RPG explores the environment using a separate EPG agent in PONG and IQN in other games. Then RPG conducts supervised learning by minimizing the hinge loss Eq (6). It is worth noting that the exploration agent (EPG or IQN) can be replaced by any existing exploration method. In our RPG implementation, we collect all trajectories with the trajectory reward no less than the threshold $c$ without eliminating the duplicated trajectories and we empirically found it is a reasonable simplification. More details of hyperparameters are provided in the Appendix Section 10.12.

**Sample-efficiency:** As the results shown in Figure 2, our approach, RPG, significantly outperform the state-of-the-art baselines in terms of sample-efficiency at all tasks. Furthermore, RPG not only achieved the most sample-efficient results, but also reached the highest final performance at ROBOTANK, DOUBLEDUNK, PITFALL, and PONG, comparing to any model-free state-of-the-art. In reinforcement learning, the stability of algorithm should be emphasized as an important issue. As we can see from the results, the performance of baselines varies from task to task. There is no single baseline consistently outperforms others. In contrast, due to the reduction from RL to supervised learning, RPG is consistently stable and effective across different environments. In addition to the stability and efficiency, RPG enjoys simplicity at the same time. In the environment PONG, it is surprising that RPG without any complicated exploration method largely surpassed the sophisticated value-function based approaches.

## 7.1 ABLATION STUDY

**The effectiveness of pairwise ranking policy and off-policy learning as supervised learning.** To get a better understanding of the underlying reasons that RPG is more sample-efficient than DQN variants, we performed ablation studies in the PONG environment by varying the combination of policy functions with the proposed off-policy learning. The results of EPG, LPG, and RPG are shown in the bottom right, Figure 2. Recall that EPG and LPG use listwise policy gradient (vanilla policy gradient using softmax as policy function) to conduct exploration, the off-policy learning minimizes the cross-entropy loss Eq (69). In contrast, RPG shares the same exploration method as EPG and LPG while uses pairwise ranking policy Eq (2) in off-policy learning that minimizes hinge loss Eq (6). We can see that RPG is more sample-efficient than EPG/LPG. We also compared the most advanced on-policy method Proximal Policy Optimization (PPO) Schulman et al. (2017) with EPG, LPG, and RPG. The proposed off-policy learning largely surpassed the best on-policy method. Therefore, we conclude that off-policy as supervised learning contributes to the sample-efficiency substantially, while pairwise ranking policy can further accelerate the learning. In addition, we compare RPG to off-policy policy gradient approaches: ACER Wang et al. (2016) and self-imitation learning Oh et al. (2018). As the results shown, the proposed off-policy learning framework is more sample-efficient than the state-of-the-art off-policy policy gradient approaches.

**The optimality-efficiency trade-off.** As reported in Figure 3, we empirically demonstrated the trade-off between the sample-efficiency and optimality, which is controlled by the trajectory reward threshold (as defined in Def 3). The higher value of trajectory reward threshold suggests we

---

[2]https://github.com/junhyukoh/self-imitation-learning
[3]Code is available at https://github.com/illidanlab/rpg.

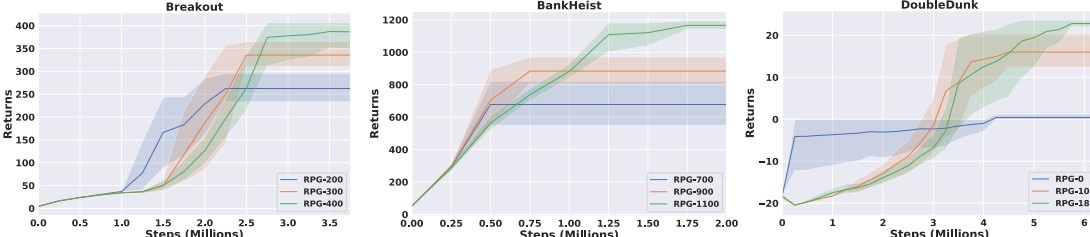

Figure 3: The trade-off between sample efficiency and optimality on DOUBLEDUNK, BREAKOUT, BANKHEIST. As the trajectory reward threshold ($c$) increase, more samples are needed for the learning to converge, while it leads to better final performance. We denote the value of $c$ by the numbers at the end of legends.

have higher requirement on defining near-optimal trajectory. This will increase the difficulty of collecting near-optimal samples during exploration, while it ensures a better final performance. These experimental results also justified that RPG is also effective in the absence of prior knowledge on trajectory reward threshold, with a mild cost on introducing an additional tuning parameter.

## 8   CONCLUSIONS

In this work, we introduced ranking policy gradient (RPG) methods that, for the first time, resolve RL problem from a ranking perspective. Furthermore, towards the sample-efficient RL, we propose an off-policy learning framework that allows RL agents to be trained in a supervised learning paradigm. The off-policy learning framework uses generalized policy iteration for exploration and exploit the stableness of supervised learning for policy learning, which accomplishes the unbiasedness, variance reduction, off-policy learning, and sample efficiency at the same time. Last but not least, empirical results show that RPG achieves superior performance as compared to the state-of-the-art.

## 9   ACKNOWLEDGEMENT

The authors would like to thank Yuan Liang, Qiaozi Gao, Rundong Zhao, Shaohua Yang, Qi Wang, Boyang Liu, Juanyuan Hong, and Mengying Sun for proofreading the earlier version of this manuscript and gave helpful suggestions on the writing. This material is based in part upon work supported by the National Science Foundation (NSF) IIS-1749940, IIS-1615597, and the Office of Naval Research (ONR) N00014-17-1-2265.

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

## 10   APPENDIX

In Table 1 we provide a brief summary of important notations used in the paper:

| Notations | Definition |
|---|---|
| $\lambda_{ij}$ | The discrepancy of the relative action value of action $i$ and action $j$. $\lambda_{ij} = \lambda_i - \lambda_j$, where $\lambda_i = \lambda(s, a_i)$. Notice that the value here is not the estimation of return, it represents which action will have relatively higher return if followed. |
| $Q^\pi(s, a)$ | The action value function or equivalently the estimation of return taking action $a$ at state $s$, following policy $\pi$. |
| $p_{ij}$ | $p_{ij} = P(\lambda_i > \lambda_j)$ denotes the probability that $i$-th action is to be ranked higher than $j$-th action. Notice that $p_{ij}$ is controlled by $\theta$ through $\lambda_i, \lambda_j$ |
| $\tau$ | A trajectory $\tau = \{s(\tau, t), a(\tau, t)\}_{t=1}^T$ collected from the environment. It is worth noting that this trajectory is not associated with any policy. It only represents a series of state-action pairs. We also use the abbreviation $s_t = s(\tau, t)$, $a_t = a(\tau, t)$. |
| $r(\tau)$ | The trajectory reward $r(\tau) = \sum_{t=1}^T r(s_t, a_t)$ is the sum of reward along one trajectory. |
| $R_{\max}$ | $R_{\max}$ is the maximum trajectory reward, i.e., $R_{\max} = \max_\tau r(\tau)$. Since we focus on MDPs with finite horizon and immediate reward, therefore the trajectory reward is bounded. |
| $M$ | $M$ is the upper bound of the absolute value trajectory reward, i.e., $|r(\tau)| \leq M, \forall \tau$. |
| $\sum_\tau$ | The summation over all possible trajectories $\tau$. |
| $p_\theta(\tau)$ | The probability of a specific trajectory is collected from the environment given policy $\pi_\theta$. $p_\theta(\tau) = p(s_0) \Pi_{t=1}^T \pi_\theta(a_t|s_t) p(s_{t+1}|s_t, a_t)$ |
| $\mathcal{T}$ | The set of all possible near-optimal trajectories. $|\mathcal{T}|$ denotes the number of near-optimal trajectories in $\mathcal{T}$. |
| $n$ | The number of training samples or equivalently state action pairs sampled from uniformly optimal policy. |
| $m$ | The number of discrete actions. |

Table 1: Notations

### 10.1   A DISCUSSION OF PRIOR WORKS ON REDUCING RL TO SL.

There are two main distinctions between supervised learning and reinforcement learning. In supervised learning, the data distribution $\mathcal{D}$ is static and training samples are assumed to be sampled i.i.d. from $\mathcal{D}$. On the contrary, the data distribution is dynamic in RL. It is determined by both environment dynamics and the learning policy. The policy keeps evolving during the learning process, which results in the dynamic data distribution in RL. Secondly, the training samples we collected are not independently distributed due to the change of learning policy. These intrinsic difficulties of RL make the learning algorithm unstable and sample-inefficient. However, if we review the state-of-the-art in RL community, every algorithm eventually acquires the policy, either explicitly or implicitly, which is a mapping from the state to an action or a probability distribution over the action space. Ultimately, there exists a supervised learning equivalent to the RL problem, if the optimal policies exist. The paradox is that it is almost impossible to construct this supervised learning equivalent on the fly, without knowing any optimal policy. Although what is the proper supervision still lingered in the RL community, pioneers have developed a set of insightful approaches to reduce RL into its SL counterpart over the past several decades. Roughly, we can classify the prior work into the following categories:

- *Expectation-Maximization (EM)* Dayan & Hinton (1997); Peters & Schaal (2007); Kober & Peters (2009); Abdolmaleki et al. (2018), etc.

- *Entropy-Regularized RL (ERL)* O'Donoghue et al. (2016); Oh et al. (2018); Haarnoja et al. (2018), etc.

- *Interactive Imitation Learning (IIL)* Daumé et al. (2009); Syed & Schapire (2010); Ross & Bagnell (2010); Ross et al. (2011); Sun et al. (2017), etc.

The early work in the EM track transfers objective by Jensen's inequality and the maximizing the lower bound of the original objective, which resembles Expectation-Maximization procedure and provides policy improvement

---

**Algorithm 1** Off-Policy Learning for Ranking Policy Gradient (RPG)

---

**Require:** The near-optimal trajectory reward threshold $c$, the number of maximal training episodes $N_{max}$.
    Maximum number of time steps in each episode $T$, and batch size $b$.

1: **while** episode $< N_{\max}$ **do**
2:    **repeat**
3:       Retrieve state $s_t$ and sample action $a_t$ by the specified exploration agent (can be random, $\epsilon$-greedy, or any RL algorithms).
4:       Collect the experience $e_t = (s_t, a_t, r_t, s_{t+1})$ and store to the replay buffer.
5:       $t = t + 1$
6:       **if** t % update step == 0 **then**
7:          Sample a batch of experience $\{e_j\}_{j=1}^b$ from the near-optimal replay buffer.
8:          Update $\pi_\theta$ based on the hinge loss Eq (6) for RPG.
9:          Update exploration agent using samples from regular replay buffer (In simple MDPs such as PONG where we access to near-optimal trajectory frequently, we can use near-optimal replay buffer to update exploration agent).
10:      **end if**
11:    **until** terminal $s_t$ or $t - t_{\text{start}} >= T$
12:    **if** return $\sum_{t=1}^T r_t \geq c$ **then**
13:       Take the near-optimal trajectory $e_t, t = 1, ..., T$ in the latest episode from the regular replay buffer into near-optimal replay buffer.
14:    **end if**
15:    **if** t % evaluation step == 0 **then**
16:       Evaluate the RPG agent by greedily choosing the action. If the best performance is reached, then stop training.
17:    **end if**
18: **end while**

---

guarantee. While pioneering at the time, these works typically focus on the simplified RL setting, such as in Dayan & Hinton (1997) the reward function is not associated with the state or in Peters & Schaal (2008) the goal is to maximize the expected immediate reward and the state distribution is assumed to be fixed. Later on in Kober & Peters (2009), the authors extended the EM framework from immediate reward into episodic return. Recent advance Abdolmaleki et al. (2018) utilizes the EM-framework on a relative entropy objective, which adds a parameter prior as regularization. As mentioned in the paper, the evaluation step using *Retrace* Munos et al. (2016) can be unstable even with linear function approximation Touati et al. (2017). In general, the estimation step in EM-based algorithms involves on-policy evaluation, which is one difficulty shared for any policy gradient methods. One of the main motivation that we want to transfer the RL into a supervised learning task is the off-policy learning enable sample efficiency.

To achieve off-policy learning, PGQ O'Donoghue et al. (2016) connected the entropy-regularized policy gradient with Q-learning under the constraint of small regularization. In the similar framework, Soft Actor-Critic Haarnoja et al. (2018) was proposed to enable sample-efficient and faster convergence under the framework of entropy-regularized RL. It is able to converge to the optimal policy that optimizes the long-term reward along with policy entropy. It is an efficient way to model the suboptimal behavior and empirically it is able to learn a reasonable policy. Although recently the discrepancy between the entropy-regularized objective and original long-term reward has been discussed in O'Donoghue (2018); Eysenbach & Levine (2019), they focus on learning stochastic policy while the proposed framework is feasible for both learning deterministic optimal policy (Corollary 1) and stochastic optimal policy (Corollary 6). In Oh et al. (2018), this work shares similarity to our work in terms of the method we collecting the samples. They collect good samples based on the past experience and then conduct the imitation learning w.r.t those good samples. However, we differentiate at how do we look at the problem theoretically. This self-imitation learning procedure was eventually connected to lower-bound-soft-Q-learning, which belongs to entropy-regularized reinforcement learning. We comment that there is a trade-off between sample-efficiency and modeling suboptimal behaviors. The more strict requirement we have on the samples collected we have less chance to hit the samples while we are more close to imitating the optimal behavior.

From the track of interactive imitation learning, initial representative works such as Ross & Bagnell (2010); Ross et al. (2011) firstly pointed out the main discrepancy between imitation learning and reinforcement learning is the i.i.d. assumption does not hold and provides SMILE Ross & Bagnell (2010) and DAGGER Ross et al. (2011) to overcome the distribution mismatch. The theorem 2.1 in Ross & Bagnell (2010) firstly analyzed if the learned policy fails to imitate the expert with a certain probability, what is the performance degradation comparing to the expert. While the theorem seems to resemble the long-term performance theorem 2, it considers the learning policy is trained through the state distribution induced by the expert, instead of state-action distribution as we did in Theorem 2. Their theorem thus may be more applicable to the situation where an interactive procedure is needed, such as querying the expert during the training process. On the contrary, we focus on directly applying

| Methods | Objective | Cont. Action | Optimality | Off-Policy | No Oracle |
|---------|-----------|--------------|------------|------------|-----------|
| EM | ✓ | ✓ | ✓ | ✗ | ✓ |
| ERL | ✗ | ✓ | ✓$^\dagger$ | ✓ | ✓ |
| IIL | ✓ | ✓ | ✓ | ✓ | ✗ |
| RPG | ✓ | ✗ | ✓ | ✓ | ✓ |

Table 2: A comparison with prior work on reducing RL to SL. The *objective* column denotes whether the goal is to maximize long-term reward. The *Cont. Action* column denotes whether the method is applicable for both continuous action space and discrete action space. The *Optimality* denotes whether the algorithms can model the optimal policy. The ✓$^\dagger$ denotes the optimality achieved by ERL is w.r.t. the entropy regularize objective instead of return. The *Off-Policy* column denotes if the algorithm enable off-policy learning. The *No Oracle* column denotes if the algorithms need to access to certain type of oracle (expert policy or expert demonstrations).

supervised learning approach without having access to the expert to label the data. The optimal state-action pairs are collected during exploration and conducting supervised learning on the replay buffer will provide a performance guarantee in terms of long-term expected reward. Concurrently, a resemble of theorem 2.1 in Ross & Bagnell (2010) is Theorem 1 in Syed & Schapire (2010), the authors reduce the apprenticeship learning to classification, under the assumption that the apprentice policy is deterministic and the misclassification rate at all time steps is bounded, which we do not make. Within the IIL track, later on the AGGREVATE Ross & Bagnell (2014) was proposed to incorporate the information of action costs to facilitate imitation learning, and a differentiable version called AGGREVATED Sun et al. (2017) was recently developed and achieved impressive empirical results. Recently, hinge loss was combined with regular $Q$-learning loss as a pre-training step for learning from demonstration Hester et al. (2018) or as a surrogate loss for imitating optimal trajectories Osa et al. (2018). In this work, we show that hinge loss constructs a new type of policy gradient method and can learn optimal policy directly.

In conclusion, our method approaches the problem of reducing RL to SL from a unique perspective that is different from all prior work. With our reformulation from RL to SL, the proposed off-policy framework preserves optimality and reduces variance simultaneously. Furthermore, it also leads to stable optimization since we are imitating a stationary target (UNOP), and it is agnostic to the knowledge of Oracle, such as expert policy or demonstrations. A multi-aspect comparison between the proposed method and relevant prior studies is summarized in Table 2.

## 10.2 RANKING POLICY GRADIENT THEOREM

The proof of Theorem 1 can be found as follows:

*Proof.* The following proof is based on direct policy differentiation Peters & Schaal (2008); Williams (1992). For concise presentation, the subscript $t$ for action value $\lambda_i, \lambda_j$, and $p_{ij}$ is omitted.

$$\nabla_\theta J(\theta) = \nabla_\theta \sum_\tau p_\theta(\tau) r(\tau)$$

$$= \sum_\tau p_\theta(\tau) \nabla_\theta \log p_\theta(\tau) r(\tau)$$

$$= \sum_\tau p_\theta(\tau) \nabla_\theta \log \left( p(s_0) \Pi_{t=1}^T \pi_\theta(a_t|s_t) p(s_{t+1}|s_t, a_t) \right) r(\tau)$$

$$= \sum_\tau p_\theta(\tau) \sum_{t=1}^T \nabla_\theta \log \pi_\theta(a_t|s_t) r(\tau)$$

$$= \mathbf{E}_{\tau \sim \pi_\theta} [\sum_{t=1}^T \nabla_\theta \log \pi_\theta(a_t|s_t) r(\tau)]$$

$$= \mathbf{E}_{\tau \sim \pi_\theta} [\sum_{t=1}^T \nabla_\theta \log(\Pi_{j=1,j\neq i}^m p_{ij}) r(\tau)]$$

$$= \mathbf{E}_{\tau \sim \pi_\theta} [\sum_{t=1}^T \nabla_\theta \sum_{j=1,j\neq i}^m \log(\frac{e^{\lambda_{ij}}}{1+e^{\lambda_{ij}}}) r(\tau)]$$

$$= \mathbf{E}_{\tau \sim \pi_\theta} [\sum_{t=1}^{T} \nabla_\theta \sum_{j=1, j \neq i}^{m} \log(\frac{1}{1 + e^{\lambda_{ji}}}) r(\tau)] \tag{7}$$

$$\approx \mathbf{E}_{\tau \sim \pi_\theta} [\sum_{t=1}^{T} \nabla_\theta \left( \sum_{j=1, j \neq i}^{m} (\lambda_i - \lambda_j)/2 \right) r(\tau)] \tag{8}$$

where the trajectory is a series of state-action pairs from $t = 1, ..., T$, $i.e. \tau = s_1, a_1, s_2, a_2, ..., s_T$. From Eq (7) to Eq (8), we use the first-order Taylor expansion of $\log(1 + e^x)|_{x=0} = \log 2 + \frac{1}{2}x + O(x^2)$ to further simplify the ranking policy gradient. □

## 10.3 DISCUSSION ON THE PROBABILITY DISTRIBUTION OF RPG

**Corollary 3.** *The pairwise ranking policy as shown in Eq (2) constructs a probability distribution over the set of actions when the action space $m$ is equal to 2, given any relative action values $\lambda_i, i = 1, 2$. For the cases with $m > 2$, this conclusion does not hold in general.*

It is easy to verify that $\pi(a_i|s) > 0$, $\sum_{i=1}^{2} \pi(a_i|s) = 1$ holds and the same conclusion cannot be applied to $m > 2$ by constructing counterexamples. However, we can introduce a dummy action $a'$ to form a probability distribution for RPG. During policy learning, the algorithm will increase the probability of best actions and the probability of dummy action will decrease. Ideally, if RPG converges to an optimal deterministic policy, the probability of taking best action is equal to one and $\pi(a'|s) = 0$. Similarly, we can introduce a dummy trajectory $\tau'$ with trajectory reward $r(\tau') = 0$ and $p_\theta(\tau') = 1 - \sum_\tau p_\theta(\tau)$. The trajectory probability forms a probability distribution since $\sum_\tau p_\theta(\tau) + p_\theta(\tau') = 1$ and $p_\theta(\tau) \geq 0 \; \forall \tau$ and $p_\theta(\tau') \geq 0$. The proof of a valid trajectory probability is similar to the following proof on $\pi(a|s)$ is a valid probability distribution with a dummy action. The practical influence of this is negligible since our goal is to increase the probability of (near-)optimal trajectories. To present in a clear way, we avoid mentioning dummy trajectory $\tau'$ in Proof 10.2 while it can be seamlessly included.

**Condition 1** (The range of $\lambda$-value). *We restrict the range of $\lambda$-values in RPG so that it satisfies $\lambda_m \geq \ln(m^{\frac{1}{m-1}} - 1)$, where $\lambda_m = \min_{i,j} \lambda_{ji}$, $m$ is the action dimension.*

This condition can be easily satisfied since in RPG we only focus on the relative relationship of $\lambda$-values and we can constrain its range so that $\lambda_m$ satisfies the condition 1. Furthermore, since we can see that $m^{\frac{1}{m-1}} > 1$ is decreasing w.r.t to action dimension $m$. The larger the action dimension, the less constraint we have on the $\lambda$-values.

**Corollary 4.** *Given Condition 1, we introduce a dummy action $a'$ and set $\pi(a = a'|s) = 1 - \sum_i \pi(a = a_i|s)$, which will construct a valid probability distribution $(\pi(a|s))$ over the action space $\mathcal{A} \cup a'$.*

*Proof.* Since we have $\pi(a = a_i|s) > 0 \; \forall i = 1, ..., m$ and $\sum_i \pi(a = a_i|s) + \pi(a = a'|s) = 1$. To prove this is a valid probability distribution, we only need to show that $\pi(a = a'|s) \geq 0$, $\forall m \geq 2$, i.e. $\sum_i \pi(a = a_i|s) \leq 1$, $\forall m \geq 2$. Let $\lambda_m = \min_{i,j} \lambda_{ji}$,

$$\sum_i \pi(a = a_i|s) \tag{9}$$

$$= \sum_i \Pi_{j=1, j \neq i}^{m} p_{ij} \tag{10}$$

$$= \sum_i \Pi_{j=1, j \neq i}^{m} \frac{1}{1 + e^{\lambda_{ji}}} \tag{11}$$

$$\leq \sum_i \Pi_{j=1, j \neq i}^{m} \frac{1}{1 + e^{\lambda_m}} \tag{12}$$

$$= m \left( \frac{1}{1 + e^{\lambda_m}} \right)^{m-1} \quad \text{use Condition 1} \tag{13}$$

$$\leq 1 \tag{14}$$

□

## 10.4 LISTWISE POLICY GRADIENT

In order to learn the stochastic policy that optimizes the ranking of actions with respect to the return, we now introduce the Listwise Policy Gradient (LPG) method. In RL, we want to optimize the probability of each action

($a_i$) to be ranked higher among all actions, which is the sum of the probabilities of all permutations such that the action $a_i$ in the top position of the list. This probability is computationally prohibitive since we need to consider the probability of $m!$ permutations. Luckily, based on Cao et al. (2007) [Theorem 6], we can model the such probability of action $a_i$ to be ranked highest given a set of relative action values by a simple softmax formulation, as described in Theorem 3.

**Theorem 3** (Theorem 6 Cao et al. (2007))*. Given the relative action values $q = [\lambda_1, ..., \lambda_m]$, the probability of action $i$ to be taken (i.e. to be ranked on the top of the list) is:*

$$\pi(a_t = a_i | s_t) = \frac{\phi(\lambda_i)}{\sum_{j=1}^m \phi(\lambda_j)} \tag{15}$$

*where $\phi(*)$ is any increasing, strictly positive function. A common choice of $\phi$ is the exponential function.*

Closely built upon the foundations from learning to rank Cao et al. (2007), we present the listwise policy gradient method, as introduced in Theorem 4.

**Theorem 4** (Listwise Policy Gradient Theorem)*. For any MDP, the gradient of the long-term reward $J(\theta) = \sum_\tau p_\theta(\tau) r(\tau)$ w.r.t. the parameter $\theta$ of listwise ranking policy takes the following form:*

$$\nabla_\theta J(\theta) = \mathbf{E}_{\tau \sim \pi_\theta} \left[ \sum_{t=1}^T \nabla_\theta \left( \log \frac{e^{\lambda_i}}{\sum_{j=1}^m e^{\lambda_j}} \right) r(\tau) \right], \tag{16}$$

*where the listwise ranking policy $\pi_\theta$ parameterized by $\theta$ is given by Eq (17) for tasks with deterministic optimal policies:*

$$a = \arg\max_i \lambda_i, \quad i = 1, \ldots, m \tag{17}$$

*or Eq (18) for stochastic optimal policies:*

$$a \sim \pi(*|s), \quad i = 1, \ldots, m \tag{18}$$

*where the policy takes the form as in Eq (19)*

$$\pi(a = a_i | s_t) = \frac{e^{\lambda_i}}{\sum_{j=1}^m e^{\lambda_j}} \tag{19}$$

*is the probability that action $i$ being ranked highest, given the current state and all the relative action values $\lambda_1 \ldots \lambda_m$.*

The proof of Theorem 4 exactly follows the direct policy differentiation Peters & Schaal (2008); Williams (1992) by replacing the policy to the form of the softmax function. The action probability $\pi(a_i|s), \forall i = 1, ..., m$ forms a probability distribution over the set of discrete actions [Cao et al. (2007) Lemma 7]. Theorem 4 states that the vanilla policy gradient Williams (1992) parameterized by a softmax layer is optimizing the probability of each action to be ranked highest, with respect to the long-term reward.

## 10.5 DISCUSSIONS ON THE OPTIMALITY PRESERVING

**Condition 2** (Initial States)*. The (near)-optimal trajectories will cover all initial states of MDP. i.e. $\{s(\tau, 1)| \forall \tau \in \mathcal{T}\} = \{s(\tau, 1)| \forall \tau\}$, where $\mathcal{T} = \{\tau | w(\tau) = 1\} = \{\tau | r(\tau) \geq c\}$.*

The Condition 2 describes what type of MDPs is directly applicable to the trajectory reward shaping (TRS, Def 3). The MDPs satisfying this condition cover a wide range of tasks such as Dialogue System Li et al. (2017), Go Silver et al. (2017), video games Bellemare et al. (2013) and all MDPs with only one initial state. If we want to preserve the optimality by TRS, the optimal trajectories of MDP needs to cover all initial states or equivalently, all initial states will lead to at least one optimal trajectory. Similarly, the near-optimality is preserved for all MDPs that its near-optimal trajectories cover all initial states.

Theoretically, it is possible to transfer more general MDPs to satisfy Condition 2 and preserve the optimality with potential-based reward shaping Ng et al. (1999). More concretely, consider the deterministic binary tree MDP ($\mathcal{M}_1$) with the set of initial states $\mathcal{S}_1 = \{s_1, s_1'\}$ as defined in Figure 4. There are eight possible trajectories in $\mathcal{M}_1$. Let $r(\tau_1) = 10 = R_{\max}, r(\tau_8) = 3, r(\tau_i) = 2, \forall i = 2, ...7$. Therefore, this MDP does not satisfy Condition 2. We can compensate the trajectory reward of best trajectory starting from $s_0'$ to the $R_{\max}$ by shaping the reward with the potential-based function $\phi(s_7') = 7$ and $\phi(s) = 0, \forall s \neq s_7'$. This reward shaping requires more prior knowledge, which may not be feasible in practice. A more realistic method is to design a dynamic trajectory reward shaping approach. In the beginning, we set $c(s) = \min_{s \in \mathcal{S}_1} r(\tau | s(\tau, 1) = s), \forall s \in \mathcal{S}_1$. Take $\mathcal{M}_1$ as an example, $c(s) = 3, \forall s \in \mathcal{S}_1$. During the exploration stage, we track the current best trajectory of each initial state and update $c(s)$ with its trajectory reward.

Nevertheless, if the Condition 2 is not satisfied, we need more sophisticated prior knowledge other than a predefined trajectory reward threshold $c$ to construct the replay buffer (training dataset of UNOP). The practical implementation of trajectory reward shaping and rigorously theoretical study for general MDPs are beyond the scope of this work.

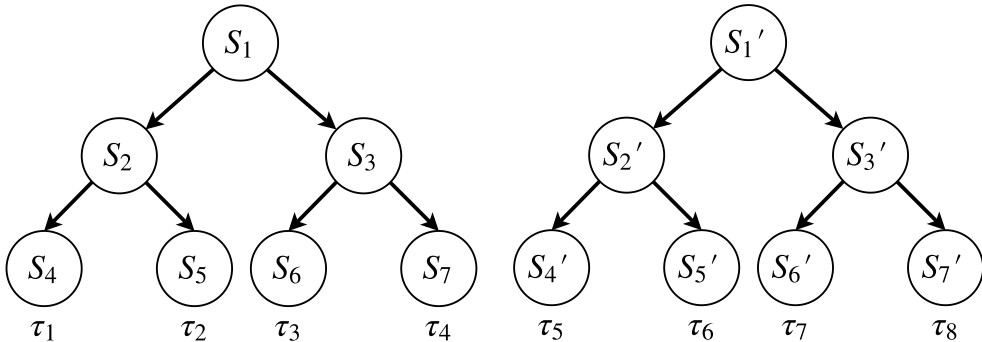

Figure 4: The binary tree structure MDP with two initial states ($\mathcal{S}_1 = \{s_1, s_1'\}$), similar as discussed in Sun et al. (2017). Each path from the root to the leaf node denotes one possible trajectory in the MDP.

## 10.6 PROOF OF LONG-TERM PERFORMANCE THEOREM 2

In this subsection, we reduce maximizing RL objective into a supervised learning problem with Theorem 2. Before that, we first prove Lemma 1 to link the log probability of a trajectory $\tau$ to its state action distribution. Then using this lemma, we can connect the trajectory probability of UNOP with its state-action distribution, from which we prove the Theorem 2.

**Lemma 1.** *Given a specific trajectory $\tau$, the averaged state-action pair log-likelihood over horizon $T$ is equal to the weighted sum over the entire state-action space, i.e.:*

$$\frac{1}{T}\sum_{t=1}^{T}\log \pi_\theta(a_t|s_t) = \sum_{s,a} p(s,a|\tau)\log \pi_\theta(a|s) \tag{20}$$

*where the sum in the right hand side is the summation over all possible state-action pairs. It is worth noting that $p(s,a|\tau)$ is not related to any policy parameters. It is the probability of a specific state-action pair $(s,a)$ in a specific trajectory $\tau$.*

*Proof.* Given a trajectory $\tau = \{(s(\tau,1), a(\tau,1)), ..., (s(\tau,T), a(\tau,T))\} = \{(s_1,a_1), ..., (s_T, a_T)\}$, denote the unique state action pairs in this trajectory as $U(\tau) = \{(s_i,a_i)\}_{i=1}^{n}$, where $n$ is the number of unique state-action pairs in $\tau$ and $n \leq T$. The number of occurrences of a state-action pair $(s_i,a_i)$ in the trajectory $\tau$ is denoted as $|(s_i,a_i)|$.

$$\frac{1}{T}\sum_{t=1}^{T}\log \pi_\theta(a_t|s_t) \tag{21}$$

$$=\sum_{i=1}^{n}\frac{|(s_i,a_i)|}{T}\log \pi_\theta(a_i|s_i) \tag{22}$$

$$=\sum_{i=1}^{n}p(s_i,a_i|\tau)\log \pi_\theta(a_i|s_i) \tag{23}$$

$$=\sum_{(s,a)\in U(\tau)}p(s,a|\tau)\log \pi_\theta(a|s) \tag{24}$$

$$=\sum_{(s,a)\in U(\tau)}p(s,a|\tau)\log \pi_\theta(a|s) + \sum_{(s,a)\notin U(\tau)}p(s,a|\tau)\log \pi_\theta(a|s) \tag{25}$$

$$=\sum_{(s,a)}p(s,a|\tau)\log \pi_\theta(a|s) \tag{26}$$

where from Eq (24) to Eq (25) we use $\sum_{(s,a)\in U(\tau)}p(s,a|\tau) = \sum_{i=1}^{n}p(s_i,a_i|\tau) = \sum_{i=1}^{n}\frac{|(s_i,a_i)|}{T} = 1$, $\therefore\ p(s,a|\tau) = 0,\ \forall(s,a)\notin U(\tau)$.

This thus completes the proof. $\square$

Now we are ready to prove the Theorem 2:

*Proof.* The following proof holds for arbitrary subset of trajectories $\mathcal{T}$ which is determined by the threshold $c$ in Def 5. The $\pi_*$ is associated with $c$ and this subset of trajectories.

$$\arg\max_\theta \sum_\tau p_\theta(\tau)w(\tau) \tag{27}$$

$$\because w(\tau) = 0, if\ \tau \notin \mathcal{T} \tag{28}$$

$$= \arg\max_\theta \frac{1}{|\mathcal{T}|} \sum_{\tau \in \mathcal{T}} p_\theta(\tau)w(\tau) \tag{29}$$

$$\text{use Lemma 3} \because p_\theta(\tau) > 0 \text{ and } w(\tau) > 0, \therefore \sum_{\tau \in \mathcal{T}} p_\theta(\tau)w(\tau) > 0 \tag{30}$$

$$= \arg\max_\theta \log(\frac{1}{|\mathcal{T}|} \sum_{\tau \in \mathcal{T}} p_\theta(\tau)w(\tau)) \tag{31}$$

$$\because \log(\sum_{i=1}^n x_i/n) \geq \sum_{i=1}^n \log(x_i)/n, \forall i, x_i > 0, \text{ we have:} \tag{32}$$

$$\log(\frac{1}{|\mathcal{T}|} \sum_{\tau \in \mathcal{T}} p_\theta(\tau)w(\tau)) \geq \sum_{\tau \in \mathcal{T}} \frac{1}{|\mathcal{T}|} \log p_\theta(\tau)w(\tau) \tag{33}$$

The lower bound holds when $p_\theta(\tau)w(\tau) = \frac{1}{|\mathcal{T}|}, \forall \tau \in \mathcal{T}$. To this end, we maximize the lower bound of the expected long-term performance.

$$\arg\max_\theta \sum_{\tau \in \mathcal{T}} \frac{1}{|\mathcal{T}|} \log p_\theta(\tau)w(\tau) \tag{34}$$

$$= \arg\max_\theta \sum_{\tau \in \mathcal{T}} \log(p(s_1)\Pi_{t=1}^T(\pi_\theta(a_t|s_t)p(s_{t+1}|s_t, a_t))w(\tau)) \tag{35}$$

$$= \arg\max_\theta \sum_{\tau \in \mathcal{T}} \log\left(p(s_1)(\Pi_{t=1}^T \pi_\theta(a_t|s_t))(\Pi_{t=1}^T p(s_{t+1}|s_t, a_t)w(\tau)\right) \tag{36}$$

$$= \arg\max_\theta \sum_{\tau \in \mathcal{T}} \left(\log p(s_1) + \sum_{t=1}^T \log p(s_{t+1}|s_t, a_t) + \sum_{t=1}^T \log \pi_\theta(a_t|s_t) + \log w(\tau)\right) \tag{37}$$

This is the reason that $w(\tau)$ can be set as arbitrary positive constant $\tag{38}$

$$= \arg\max_\theta \frac{1}{|\mathcal{T}|} \sum_{\tau \in \mathcal{T}} \sum_{t=1}^T \log \pi_\theta(a_t|s_t) \tag{39}$$

$$= \arg\max_\theta \frac{1}{|\mathcal{T}|T} \sum_{\tau \in \mathcal{T}} \sum_{t=1}^T \log \pi_\theta(a_t|s_t) \tag{40}$$

$$= \arg\max_\theta \frac{1}{|\mathcal{T}|} \sum_{\tau \in \mathcal{T}} \frac{1}{T} \sum_{t=1}^T \log \pi_\theta(a_t|s_t) \text{ Use Assumption 2 the existence of UNOP.} \tag{41}$$

$$= \arg\max_\theta \sum_{\tau \in \mathcal{T}} p_{\pi_*}(\tau) \frac{1}{T} \left(\sum_{t=1}^T \log \pi_\theta(a_t|s_t)\right) \tag{42}$$

where $\pi_*$ is a UNOP (Def 5). $\tag{43}$

$$\therefore p_{\pi_*}(\tau) = 0 \forall \tau \notin \mathcal{T} \tag{44}$$

Eq (44) can be established based on $\sum_{\tau \in \mathcal{T}} p_{\pi_*}(\tau) = \sum_{\tau \in \mathcal{T}} 1/|\mathcal{T}| = 1$ $\tag{45}$

$$= \arg\max_\theta \sum_\tau p_{\pi_*}(\tau) \frac{1}{T} \left(\sum_{t=1}^T \log \pi_\theta(a_t|s_t)\right) \text{ Use lemma 1}$$

$$= \arg\max_\theta \sum_\tau p_{\pi_*}(\tau) \sum_{s,a} p(s, a|\tau) \log \pi_\theta(a|s) \tag{46}$$

The 2nd sum is over all possible state-action pairs. $(s, a)$ represents a specific state-action pair.

$$= \arg\max_\theta \sum_\tau \sum_{s,a} p_{\pi_*}(\tau)p(s, a|\tau) \log \pi_\theta(a|s) \tag{47}$$

$$= \arg\max_\theta \sum_{s,a} \sum_\tau p_{\pi_*}(\tau)p(s, a|\tau) \log \pi_\theta(a|s) \tag{48}$$

$$= \arg \max_{\theta} \sum_{s,a} p_{\pi_*}(s,a) \log \pi_{\theta}(a|s) \tag{49}$$

In this proof we use $s_t = s(\tau, t)$ and $a_t = a(\tau, t)$ as abbreviations, which denote the $t$-th state and action in the trajectory $\tau$, respectively. $|\mathcal{T}|$ denotes the number of trajectories in $\mathcal{T}$. We also use the definition of $w(\tau)$ to only focus on near-optimal trajectories. We set $w(\tau) = 1$ for simplicity but it will not affect the conclusion if set to other constants.

**Optimality:** Furthermore, the optimal solution for the objective function Eq (49) is a uniformly (near)-optimal policy $\pi_*$.

$$\arg \max_{\theta} \sum_{s,a} p_{\pi_*}(s,a) \log \pi_{\theta}(a|s) \tag{50}$$

$$= \arg \max_{\theta} \sum_{s} p_{\pi_*}(s) \sum_{a} \pi_*(a|s) \log \pi_{\theta}(a|s) \tag{51}$$

$$= \arg \max_{\theta} \sum_{s} p_{\pi_*}(s) \sum_{a} \pi_*(a|s) \log \pi_{\theta}(a|s) - \sum_{s} p_{\pi_*}(s) \sum_{a} \log \pi_*(a|s) \tag{52}$$

$$= \arg \max_{\theta} \sum_{s} p_{\pi_*}(s) \sum_{a} \pi_*(a|s) \log \frac{\pi_{\theta}(a|s)}{\pi_*(a|s)} \tag{53}$$

$$= \arg \max_{\theta} \sum_{s} p_{\pi_*}(s) \sum_{a} -KL(\pi_*(a|s)||\pi_{\theta}(a|s)) = \pi_* \tag{54}$$

Therefore, the optimal solution of Eq (49) is also the (near)-optimal solution for the original RL problem since $\sum_{\tau} p_{\pi_*}(\tau) r(\tau) = \sum_{\tau \in \mathcal{T}} \frac{1}{|\mathcal{T}|} r(\tau) \geq c = R_{\max} - \epsilon$. The optimal solution is obtained when we set $c = R_{\max}$. □

**Lemma 2.** *Given any optimal policy $\pi$ of MDP satisfying Condition 2, $\forall \tau \notin \mathcal{T}$, we have $p_{\pi}(\tau) = 0$, where $\mathcal{T}$ denotes the set of all possible optimal trajectories in this lemma. If $\exists \tau \notin \mathcal{T}$, such that $p_{\pi}(\tau) > 0$, then $\pi$ is not optimal policy.*

*Proof.* We prove this by contradiction. We assume $\pi$ is an optimal policy. If $\exists \tau' \notin \mathcal{T}$, such that 1) $p_{\pi}(\tau') \neq 0$, or equivalently: $p_{\pi}(\tau') > 0$ since $p_{\pi}(\tau') \in [1, 0]$. and 2) $\tau' \notin \mathcal{T}$. We can find a better policy $\pi'$ by satisfying the following three conditions:

$$p_{\pi'}(\tau') = 0 \text{ and}$$
$$p_{\pi'}(\tau_1) = p_{\pi}(\tau_1) + p_{\pi}(\tau'), \tau_1 \in \mathcal{T} \text{ and}$$
$$p_{\pi'}(\tau) = p_{\pi}(\tau), \forall \tau \notin \{\tau', \tau_1\}$$

Since $p_{\pi'}(\tau) \geq 0, \forall \tau$ and $\sum_{\tau} p_{\pi'}(\tau) = 1$, therefore $p_{\pi'}$ constructs a valid probability distribution. Then the expected long-term performance of $\pi'$ is greater than that of $\pi$:

$$\sum_{\tau} p_{\pi'}(\tau) w(\tau) - \sum_{\tau} p_{\pi}(\tau) w(\tau)$$
$$= \sum_{\tau \notin \{\tau', \tau_1\}} p_{\pi'}(\tau) w(\tau) + p_{\pi'}(\tau_1) * w(\tau_1) + p_{\pi'}(\tau') * w(\tau')$$
$$- (\sum_{\tau \notin \{\tau', \tau_1\}} p_{\pi}(\tau) w(\tau) + p_{\pi}(\tau_1) * w(\tau_1) + p_{\pi}(\tau') * w(\tau'))$$
$$= p_{\pi'}(\tau_1) * w(\tau_1) + p_{\pi'}(\tau') * w(\tau') - (p_{\pi}(\tau_1) * w(\tau_1) + p_{\pi}(\tau') * w(\tau'))$$
$$\because \tau' \notin \mathcal{T}, \therefore w(\tau') = 0 \text{ and } \tau_1 \in \mathcal{T}, \therefore w(\tau) = 1$$
$$= p_{\pi'}(\tau_1) - p_{\pi}(\tau_1)$$
$$= p_{\pi}(\tau_1) + p_{\pi}(\tau') - p_{\pi}(\tau_1) = p_{\pi}(\tau') > 0$$

Essentially, we can find a policy $\pi'$ that has higher probability on the optimal trajectory $\tau_1$ and zero probability on $\tau'$. This indicates that it is a better policy than $\pi$. Therefore, $\pi$ is not an optimal policy and it contradicts our assumption, which proves that such $\tau'$ does not exist. Therefore, $\forall \tau \notin \mathcal{T}$, we have $p_{\pi}(\tau) = 0$. □

**Lemma 3** (Policy Performance). *$\forall \tau$, $p_{\theta}(\tau) > 0$, if the policy takes the form as in Eq (15) or Eq (2). This means for all possible trajectories allowed by the environment, the policy takes the form of either ranking policy or softmax will generate this trajectory with probability $p_{\theta}(\tau) > 0$. It is worth noting that because of this property, $\pi_{\theta}$ is not an optimal policy according to Lemma 2, though it can be arbitrarily close to the optimal policy.*

*Proof.*

$$\because p(\tau) = p(s_1)\Pi_{t=1}^{T}(\pi_\theta(a_t|s_t)p(s_{t+1}|s_t,a_t)) \tag{55}$$

$$\text{and } \pi_\theta(a_t|s_t) > 0, p(s_1) > 0, p(s_{t+1}|s_t,a_t) > 0. \tag{56}$$

$$if\ p(s_{t+1}|s_t,a_t) = 0 \text{ or } p(s_1) = 0,$$

then the probability of sampling $\tau$ from any policy is zero. This trajectory does not exist.

$$\therefore p_\theta(\tau) > 0. \tag{57}$$

This thus completes the proof. □

## 10.7 PROOF OF PERFORMANCE POLICY GRADIENT COROLLARIES

**Corollary 5** (Ranking performance policy gradient). *Optimizing the lower bound of expected long-term performance (defined in Eq (4)) using pairwise ranking policy (Eq (2)) can be approximately optimized by the following loss:*

$$\min_\theta \sum_{s,a_i} p_{\pi_*}(s,a_i)\left(\sum_{j=1,j\neq i}^{m} \max(0, margin + \lambda(s,a_j) - \lambda(s,a_i))\right), \tag{58}$$

*where margin is a small positive value. We set margin equal to one in our experiments.*

*Proof.* In RPG, the policy $\pi_\theta(a|s)$ is defined as in Eq (2). We then replace the action probability distribution in Eq (5) with the RPG policy.

$$\because \pi(a = a_i|s) = \Pi_{j=1,j\neq i}^{m} p_{ij} \tag{59}$$

Because RPG is fitting a deterministic optimal policy,

we denote the optimal action given sate $s$ as $a_i$, then we have

$$\max_\theta \sum_{s,a_i} p_{\pi_*}(s,a_i)\log \pi(a_i|s) \tag{60}$$

$$= \max_\theta \sum_{s,a_i} p_{\pi_*}(s,a_i)\log(\Pi_{j\neq i,j=1}^{m} p_{ij}) \tag{61}$$

$$= \max_\theta \sum_{s,a_i} p_{\pi_*}(s,a_i)\log \Pi_{j\neq i,j=1}^{m} \frac{1}{1+e^{\lambda_{ji}}} \tag{62}$$

$$= \min_\theta \sum_{s,a_i} p_{\pi_*}(s,a_i)\sum_{j\neq i,j=1}^{m} \log(1+e^{\lambda_{ji}}) \text{ first order Taylor expansion} \tag{63}$$

$$\approx \min_\theta \sum_{s,a_i} p_{\pi_*}(s,a_i)\sum_{j\neq i,j=1}^{m} \lambda_{ji} \qquad \text{s.t. } |\lambda_{ij}| = c < 1, \forall i,j,s \tag{64}$$

$$= \min_\theta \sum_{s,a_i} p_{\pi_*}(s,a_i)\sum_{j\neq i,j=1}^{m} (\lambda_j - \lambda_i) \qquad \text{s.t. } |\lambda_i - \lambda_j| = c < 1, \forall i,j,s \tag{65}$$

$$\Rightarrow \min_\theta \sum_{s,a_i} p_{\pi_*}(s,a_i)L(s_i,a_i) \tag{66}$$

where the pairwise loss $L(s,a_i)$ is defined as:

$$\mathcal{L}(s,a_i) = \sum_{j=1,j\neq i}^{|A|} \max(0, \text{margin} + \lambda(s,a_j) - \lambda(s,a_i)), \tag{67}$$

where the margin in Eq (66) is a small positive constant. $\tag{68}$

From Eq (65) to Eq (66), we consider learning a deterministic optimal policy $a_i = \pi^*(s)$, where we use index $i$ to denote the optimal action at each state. The optimal $\lambda$-values minimizing Eq (65) (denoted by $\lambda^1$) need to satisfy $\lambda_i^1 = \lambda_j^1 + c, \forall j \neq i$. The optiaml $\lambda$-values minimizing Eq (66) (denoted by $\lambda^2$) need to satisfy $\lambda_i^2 = \max_{j\neq i} \lambda_j^2 + \text{margin}, \forall j \neq i$. In both cases, the optimal policies from solving Eq (65) and Eq (66) are the same: $\pi(s) = \arg\max_k \lambda_k^1 = \arg\max_k \lambda_k^2 = a_i$. Therefore, we use Eq (66) as a surrogate optimization problem of Eq (65). □

### 10.7.1 LISTWISE PERFORMANCE POLICY GRADIENT

**Corollary 6** (Listwise performance policy gradient). *Optimizing the lower bound of expected long-term performance by the listwise ranking policy (Eq (19)) is equivalent to:*

$$\max_{\theta} \sum_s p_{\pi_*}(s) \sum_{i=1}^m \pi_*(a_i|s) \log \frac{e^{\lambda_i}}{\sum_{j=1}^m e^{\lambda_j}} \tag{69}$$

The proof of Corollary 6 is a direct application of theorem 2 by replacing policy with the softmax function.

### 10.8 POLICY GRADIENT VARIANCE REDUCTION

**Corollary 7** (Variance reduction). *Given a stationary policy, , the upper bound of the variance of each dimension of policy gradient is $O(T^2 C^2 M^2)$. The upper bound of gradient variance of maximizing the lower bound of long-term performance Eq (5) is $O(C^2)$, where $C$ is the maximum norm of log gradient based on Assumption 3. The upper bound of gradient variance by supervised learning compared to that of the regular policy gradient is reduced by an order of $O(T^2 M^2)$, given $M > 1, T > 1$, which is a very common situation in practice.*

*Proof.* The regular policy gradient of policy $\pi_\theta$ is given as Williams (1992):

$$\sum_\tau p_\theta(\tau)[\sum_{t=1}^T \nabla_\theta \log(\pi_\theta(a(\tau,t)|s(\tau,t)))r(\tau)] \tag{70}$$

The regular policy gradient variance of the $i$-th dimension is denoted as follows:

$$Var\left(\sum_{t=1}^T \nabla_\theta \log(\pi_\theta(a(\tau,t)|s(\tau,t))_i)r(\tau)\right) \tag{71}$$

We denote $x_i(\tau) = \sum_{t=1}^T \nabla_\theta \log(\pi_\theta(a(\tau,t)|s(\tau,t))_i)r(\tau)$ for convenience. Therefore, $x_i$ is a random variable. Then apply $var(x) = \mathbf{E}_{p_\theta(\tau)}[x^2] - \mathbf{E}_{p_\theta(\tau)}[x]^2$, we have:

$$Var\left(\sum_{t=1}^T \nabla_\theta \log(\pi_\theta(a(\tau,t)|s(\tau,t))_i)r(\tau)\right) \tag{72}$$

$$=Var\left(x_i(\tau)\right) \tag{73}$$

$$=\sum_\tau p_\theta(\tau)x_i(\tau)^2 - [\sum_\tau p_\theta(\tau)x_i(\tau)]^2 \tag{74}$$

$$\leq \sum_\tau p_\theta(\tau)x_i(\tau)^2 \tag{75}$$

$$=\sum_\tau p_\theta(\tau)[\sum_{t=1}^T \nabla_\theta \log(\pi_\theta(a(\tau,t)|s(\tau,t))_i)r(\tau)]^2 \quad (\text{use } M \geq |r(\tau)|, \forall \tau) \tag{76}$$

$$\leq \sum_\tau p_\theta(\tau)[\sum_{t=1}^T \nabla_\theta \log(\pi_\theta(a(\tau,t)|s(\tau,t))_i)]^2 M^2 \tag{77}$$

$$=M^2 \sum_\tau p_\theta(\tau)[\sum_{t=1}^T \sum_{k=1}^T \nabla_\theta \log(\pi_\theta(a(\tau,t)|s(\tau,t))_i)\nabla_\theta \log(\pi_\theta(a(\tau,k)|s(\tau,k)_i)] \ (\text{Assumption 3}) \tag{78}$$

$$\leq M^2 \sum_\tau p_\theta(\tau)[\sum_{t=1}^T \sum_{k=1}^T C^2] \tag{79}$$

$$=M^2 \sum_\tau p_\theta(\tau)T^2 C^2 \tag{80}$$

$$=T^2 C^2 M^2 \tag{81}$$

The policy gradient of long-term performance (Def 4)

$$\sum_{s,a} p_{\pi_*}(s,a)\nabla_\theta \log \pi_\theta(a|s) \tag{82}$$

The policy gradient variance of the $i$-th dimension is denoted as

$$var(\nabla_\theta \log \pi_\theta(a|s)_i) \tag{83}$$

Then the upper bound is given by

$$var(\nabla_\theta \log \pi_\theta(a|s)_i) \tag{84}$$

$$= \sum_{s,a} p_{\pi_*}(s,a)[\nabla_\theta \log \pi_\theta(a|s)_i]^2 - \left[\sum_{s,a} p_{\pi_*}(s,a)\nabla_\theta \log \pi_\theta(a|s)_i\right]^2 \tag{85}$$

$$\leq \sum_{s,a} p_{\pi_*}(s,a)[\nabla_\theta \log \pi_\theta(a|s)_i]^2 \text{ use Assumption 3} \tag{86}$$

$$\leq \sum_{s,a} p_{\pi_*}(s,a)C^2 \tag{87}$$

$$= C^2 \tag{88}$$

This thus completes the proof. $\qquad \square$

## 10.9   DISCUSSIONS OF ASSUMPTION 2

In this section, we show that UNOP exists in a range of MDPs. Notice that the lemma 4 shows the sufficient conditions of satisfying Asumption 2 rather than necessary conditions.

**Lemma 4.** *For MDPs defined in Section 3 satisfying the following conditions:*

- *Each initial state leads to one optimal trajectory. This also indicates $|\mathcal{S}_1| = |\mathcal{T}|$, where $\mathcal{T}$ denotes the set of optimal trajectories in this lemma, $\mathcal{S}_1$ denotes the set of initial states.*

- *Deterministic transitions, i.e., $p(s'|s,a) \in \{0,1\}$.*

- *Uniform initial state distribution, i.e., $p(s_1) = \frac{1}{|\mathcal{T}|}, \forall s_1 \in \mathcal{S}_1$.*

*Then we have: $\exists \pi_*$, where s.t. $p_{\pi_*}(\tau) = \frac{1}{|\mathcal{T}|}, \forall \tau \in \mathcal{T}$. It means that a deterministic uniformly optimal policy always exists for this MDP.*

*Proof.* We can prove this by construction. The following analysis applies for any $\tau \in \mathcal{T}$.

$$p_{\pi_*}(\tau) = \frac{1}{|\mathcal{T}|} \tag{89}$$

$$\Longleftrightarrow \log p_{\pi_*}(\tau) = -\log |\mathcal{T}| \tag{90}$$

$$\Longleftrightarrow \log p(s_1) + \sum_{t=1}^{T} \log p(s_{t+1}|s_t, a_t) + \sum_{t=1}^{T} \log \pi_*(a_t|s_t) = -\log |\mathcal{T}| \tag{91}$$

$$\Longleftrightarrow \sum_{t=1}^{T} \log \pi_*(a_t|s_t) = -\log p(s_1) - \sum_{t=1}^{T} \log p(s_{t+1}|s_t, a_t) - \log |\mathcal{T}| \tag{92}$$

where we use $a_t, s_t$ as abbreviations of $a(\tau,t), s(\tau,t)$.

We denote $D(\tau) = -\log p(s_1) - \sum_{t=1}^{T} \log p(s_{t+1}|s_t, a_t) > 0$

$$\Longleftrightarrow \sum_{t=1}^{T} \log \pi_*(a_t|s_t) = D(\tau) - \log |\mathcal{T}| \tag{93}$$

$\therefore$ we can obtain a uniformly optimal policy by solving the nonlinear programming:

$$\sum_{t=1}^{T} \log \pi_*(a(\tau,t)|s(\tau,t)) = D(\tau) - \log |\mathcal{T}| \ \forall \tau \in \mathcal{T} \tag{94}$$

$$\log \pi_*(a(\tau,t)|s(\tau,t)) = 0, \ \forall \tau \in \mathcal{T}, t = 1, ..., T \tag{95}$$

$$\sum_{i=1}^{m} \pi_*(a_i|s(\tau,t)) = 1, \ \forall \tau \in \mathcal{T}, t = 1, ..., T \tag{96}$$

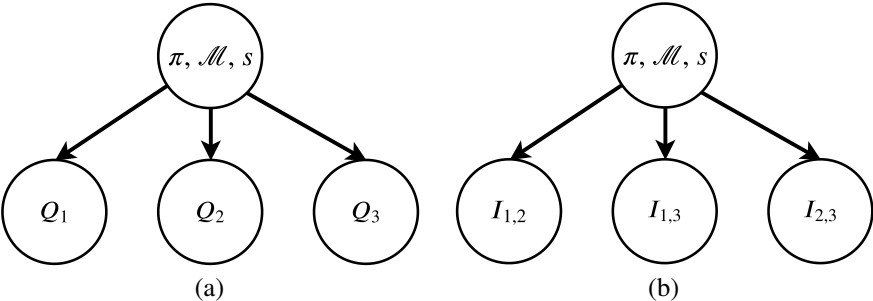

Figure 5: The directed graph that describes the conditional independence of pairwise relationship of actions, where $Q_1$ denotes the return of taking action $a_1$ at state $s$, following policy $\pi$ in $\mathcal{M}$, i.e., $Q_{\mathcal{M}}^{\pi}(s, a_1)$. $I_{1,2}$ is a random variable that denotes the pairwise relationship of $Q_1$ and $Q_2$, i.e., $I_{1,2} = 1$, i.i.f. $Q_1 \geq Q_2$, o.w. $I_{1,2} = 0$.

Use the condition $p(s_1) = \frac{1}{|\mathcal{T}|}$, then we have:

$$\because \sum_{t=1}^{T} \log \pi_*(a(\tau, t)|s(\tau, t)) = \sum_{t=1}^{T} \log 1 = 0 \ (\text{LHS of } Eq \ (94)) \tag{97}$$

$$\because -\log p(s_1) - \sum_{t=1}^{T} \log p(s_{t+1}|s_t, a_t) - \log |\mathcal{T}| = \log |\mathcal{T}| - 0 - \log |\mathcal{T}| = 0 \ (\text{RHS of } Eq \ (94)) \tag{98}$$

$$\therefore D(\tau) - \log |\mathcal{T}| = \sum_{t=1}^{T} \log \pi_*(a(\tau, t)|s(\tau, t)), \ \forall \tau \in \mathcal{T}. \tag{99}$$

Also the deterministic optimal policy satisfies the conditions in Eq (95 96). Therefore, the deterministic optimal policy is a uniformly optimal policy. This lemma describes one type of MDP in which UOP exists. From the above reasoning, we can see that as long as the system of non-linear equations Eq (94 95 96) has a solution, the uniformly (near-)optimal policy exists. □

**Lemma 5** (Hit optimal trajectory). *The probability that a specific optimal trajectory was not encountered given an arbitrary softmax policy $\pi_\theta$ is exponentially decreasing with respect to the number of training episodes. No matter a MDP has deterministic or probabilistic dynamics.*

*Proof.* Given a specific optimal trajectory $\tau = \{s(\tau, t), a(\tau, t)\}_{t=1}^{T}$, and an arbitrary stationary policy $\pi_\theta$, the probability that has never encountered at the $n$-th episode is $[1 - p_\theta(\tau)]^n = \xi^n$, based on lemma 3, we have $p_\theta(\tau) > 0$, therefore we have $\xi \in [0, 1)$. □

## 10.10 Discussions of Assumption 1

Intuitively, given a state and a stationary policy $\pi$, the relative relationships among actions can be independent, considering a fixed MDP $\mathcal{M}$. The relative relationship among actions is the relative relationship of actions' return. Starting from the same state, following a stationary policy, the actions' return is determined by MDP properties such as environment dynamics, reward function, etc.

More concretely, we consider a MDP with three actions $(a_1, a_2, a_3)$ for each state. The action value $Q_{\mathcal{M}}^{\pi}$ satisfies the Bellman equation in Eq (100). Notice that in this subsection, we use $Q_{\mathcal{M}}^{\pi}$ to denote the action value that estimates the absolute value of return in $\mathcal{M}$.

$$Q_{\mathcal{M}}^{\pi}(s, a_i) = r(s, a_i) + \max_a \mathbf{E}_{s' \sim p(*|s,a)} Q_{\mathcal{M}}^{\pi}(s', a), \forall i = 1, 2, 3. \tag{100}$$

As we can see from Eq (100), $Q_{\mathcal{M}}^{\pi}(s, a_i), i = 1, 2, 3$ is only related to $s, \pi$, and environment dynamics $\mathbf{P}$. It means if $\pi$, $\mathcal{M}$ and $s$ are given, the action values of three actions are determined. Therefore, we can use a directed graph Bishop (2006) to model the relationship of action values, as shown in Figure 5 (a). Similarly, if we only consider the ranking of actions, this ranking is consistent with the relationship of actions' return, which is also determined by $s, \pi$, and $\mathbf{P}$. Therefore, the pairwise relationship among actions can be described as the directed graph in Figure 5 (b), which establishes the conditional independence of actions' pairwise relationship. Based on the above reasoning, we conclude that Assumption 1 is realistic.

## 10.11 EXPLORATION EFFICIENCY

The proposed off-policy learning framework indicates the sample complexity is related to exploration efficiency and supervised learning efficiency. Given a specific MDP, the exploration efficiency of an exploration strategy can be quantified by how frequently we can encounter different (near)-optimal trajectories in the first $k$ episodes. The supervised learning efficiency under the probably approximately correct framework Valiant (1984) is how many samples we need to collect so that we can achieve good generalization performance with high probability. Jointly consider the efficiency in two stages, we can theoretically analyze the sample complexity of the proposed off-policy learning framework, which will be provided in the long version of this work.

Improving exploration efficiency is not the focus of this work. In general, exploration efficiency is highly related to the properties of MDP, such as transition probabilities, horizon, action dimension, etc. The exploration strategy should be designed according to certain domain knowledge of the MDP to improve the efficiency. Therefore, we did not specify our exploration strategy but adopt the state-of-the-art to conduct exploration.

Based on the above discussion, we can see that how frequently we can encounter different (near)-optimal trajectories is a bottleneck of sample efficiency for RPG. In the MDPs with small the transition probabilities of reaching the near-optimal trajectories, we rarely collect any near-optimal trajectories during the early stage of exploration. The benefit of applying the proposed off-policy framework would be limited.

## 10.12 HYPERPARAMETERS

We present the training details of ranking policy gradient in Table 3. The network architecture is the same as the convolution neural network used in DQN Mnih et al. (2015). We update the RPG network every four timesteps with a minibatch of size 32. The replay ratio is equal to eight for all baselines and RPG (except for ACER we use the default setting in openai baselines Dhariwal et al. (2017) for better performance).

Table 3: Hyperparameters of RPG network

| Hyperparameters | Value |
|---|---|
| Architecture | Conv(32-8×8-4) |
| | -Conv(64-4×4-2) |
| | -Conv(64-3×3-2) |
| | -FC(512) |
| Learning rate | 0.0000625 |
| Batch size | 32 |
| Replay buffer size | 1000000 |
| Update period | 4 |
| Margin in Eq (6) | 1 |

