# OpenReview forum: "Ranking Policy Gradient"
_ICLR.cc/2020/Conference — Accept (Poster)_

### Official Review · AnonReviewer2 · 2019-10-21
**Official Blind Review #2**

**Rating:** 6

**Review:**

This paper proposes to reparameterize the policy using a form of ranking, and updating the policy gradient update accordingly. This on it's own isn't particularly useful, however the authors use this insight to attempt to convert the RL problem into a supervised learning problem. I think this is an interesting paper and approach, however some issues remain.

High level:

The reward threshold appears to require a-priori knowledge of the correct scale to set it at, and the maximum possible reward, which is unfortunate. Also I'm not sure it would make sense in a sparse reward environment, or an environment with small reward leading up to a single large reward, since the trajectories that get small reward might be discarded erroneously. Maybe there should be an annealing schedule for this parameter? This deserves some discussion at least.

The experimental results are very good, however you are using off-policy replay data and therefore have the issue that using the same data many times can (often) improve performance for any off-policy algorithm. This needs to be controlled and investigated very closely, however I don't even see the batch size being used listed anywhere. The authors need to really demonstrate that the benefit is coming from their algorithm and *not* just the re-use of replay data that the baselines don't get to see as much of.

More minor:

Firstly, there is very bad writing in places, .e.g, abstract: "The state-of-the-art uses action value function to derive policy". Things like this appear in many places.

Q-learning, Q-values sometimes Q is upper case sometimes lower case.

Relative action values are never defined, but if they are what I think they are then they are usually referred to as 'advantage values', and are generally given the notation of A(s,a).

Does the policy in (2) sum to 1?

Assumption 1 is written very confusingly. I would state it as something like:

"Assumption 1. For a state s and action i the set of events {E^i_j}_{j \neq i}, where E^i_j corresponds to action ai being ranked higher than action aj, are conditionally independent, given a MDP and a stationary policy."

Assumption 1: What is is conditioned on?

Thm 1: the statement of the theorem has an equality, but the proof has a clear 'approximate equality', this is very misleading since the statement is not actually true!

I don't follow the proof of corollary 5, especially the sentence explaining (65) -> (66), which doesn't parse. Again the proof is actually approximate but the statement is given using an equality, which is false.

The appendix is very messy and several parts of it just consist of series of equations with no supporting explanation.

Refs:
The original DQN paper (Mnih et al) should be cited after you mention DQN.

I was surprised not to see references to
ACER: https://arxiv.org/abs/1611.01224 and
PGQ: https://arxiv.org/abs/1611.01626 when discussing the off-policy / RL + SL work.

Further, in your discussion about entropy regularized RL "However, the discrepancy between
the entropy-regularized objective and original long-term reward exists due to the entropy term." - this has been corrected by a recent work: https://arxiv.org/abs/1807.09647. Probably worth mentioning it here too.

**Experience Assessment:**

I have published in this field for several years.

**Review Assessment: Checking Correctness Of Derivations And Theory:**

I did not assess the derivations or theory.

**Review Assessment: Checking Correctness Of Experiments:**

I assessed the sensibility of the experiments.

**Review Assessment: Thoroughness In Paper Reading:**

I read the paper at least twice and used my best judgement in assessing the paper.

---

> ### Author Response · Authors · 2019-11-11
> **Response to Reviewer #2**
>
> Thank you for doing a detailed review of our work and providing insightful comments. We have worked to incorporate some of suggests into the paper to improve the clarity:
>
> Q1: The sparse reward task:
> The sparse reward tasks you mentioned belong to one type of tasks leading to very low
> frequency of hitting different near-optimal trajectories, which is closely related to our discussion on exploration efficiency in section 9.11. The sample complexity of the proposed method depends on this exploration efficiency. When applying the proposed off-policy framework to sparse reward tasks, we typically have a low probability of hitting different near-optimal trajectories, and thus the framework will not be improved significantly. We added more detailed discussion on this in section 9.11.
>
> Q2: The reward threshold C:
> 1). The benefit of C:
> Empirically, it provides a controllable handle to trade off the efficiency and optimality in practice (Figure 3). Theoretically, C leads to a stationary target (UNOP) for stable training, variance reduction, and optimality preserving.
> 2). Prior knowledge similar to C is also used in many other approaches.
> Many existing works clip the reward to [-1, 1] (e.g., dopamine) or normalize the reward to stabilize the training (e.g. openai baselines). In Self imitation learning, the trajectories with positive rewards are selected as the initial candidates for good trajectories (see self-imitation-learning/blob/master/baselines/common/self_imitation.py L269 ). These implementations depend on this domain knowledge on reward structure. At the end of section 5, we also give tasks where the reward structure is known. Therefore we believe that the assumption of knowing C is still reasonable. Furthermore, we can treat C as an additional tuning parameter if C is unknown.
>
> Q3: “Demonstrate that the benefit is coming from their algorithm and not just the re-use of replay data that the baselines don't get to see as much of.”
> We use the same batch size and same update frequency as the baselines, which means the same number of samples were fed into the network for updating (same replay ratio). We did not increase the frequency of reusing the data. We have added a more detailed description of the hyperparameter setting in section 9.12. Given the proposed method and the baselines are trained with the same replay ratio, the results show that the improvement of sample-efficiency comes from the proposed algorithm.
>
> Q4: What is the definition of relative action values?
> The relative action value denotes the relative order of actions. The difference between relative action values and the advantage function A(s, a) = Q(s, a) – V(s) is that relative action value only represents relative order. Its magnitude has no quantitative relationship with the return and the state-value function. We add definition 1 and remark 1 in Section 4 to clarify this.
>
> Q5: Does the policy in (2) sum to 1?
> In the case where we only have two actions, the policy in (2) sums up to 1. In the cases where we have more than two actions, the sum of the policy probabilities in (2) is smaller than or equal to 1. Therefore, we can introduce a dummy action to complete a valid probability distribution. For a more detailed discussion, please refer to Section 9.3 in the appendix.
>
> Q6: Assumption 1.
> Thanks for improving the writing of Assumption 1. Assumption 1 is conditioned on the MDP and a stationary policy. We also discussed this in Appendix Section 9.10 showing why this is a reasonable assumption.
>
> Q7: The proof of corollary 5, (65) -> (66)
> This will be more clear if the definition of lambda values is clarified. We add a more detailed explanation at the end of the proof in Section 9.7. In short, the lambda values obtained from solving (63) and (65) will result in the same optimal deterministic policy.
>
> Q8: References and Appendix.
> We thank the reviewer for pointing out related references. For ACER we have cited in the related work (Sec 2) and we added it as an extra baseline in the ablation study. We also added the discussion of PGQ and K-learning in Section 9.1.

---

### Official Review · AnonReviewer3 · 2019-10-21
**Official Blind Review #3**

**Rating:** 3

**Review:**

This paper presents a new view on policy gradient methods from the perspective of ranking. The end goal in policy learning is to achieve the right ranking of actions at a state (in the case when deterministic policies are optimal), and the paper proposes a method of doing this inspired from the work on learning to rank. They further argue that in the case with stochastic optimal policies, REINFORCE with softmax policies is rank wise optimal, which is not surprising, but at the same time interesting as well. The other main part of the paper is casting off-policy RL as supervised learning similar to work on self-imitation learning and reward weighted regression methods. This section is presented differently from the past analyses of self-imitation methods and requires the existence of UNOP, which seems like a strong assumption. They then instantiate the framework with GPI based exploration, and show that it achieves better performance than IQN and Rainbow on a subset of atari games.

I am leaning towards a weak reject for this paper, although I am happy to revise my score based on the rebuttal. While the paper is interesting with regards to the ranking perspective, I am not fully convinced about the novelty of the reduction of off-policy learning to supervised learning. This appears already in past works (which the paper cites in the appendix) and the assumption of the existence of UNOP seems strong. I find using a Q-learning agent for exploration a bit complicated and perhaps unnecessary. Also, the paper currently lacks intuition about the effectiveness of their policy gradient approach on top of the data collected from an DQN-based agent. Since reward shaping is done at the trajectory level, why would we expect the supervised regression step to do better than the best trajectory in the data? Also, can the same Q-learning based method perform better if one controls for the number of gradient updates? How would the other methods such as self-imitation or reward weighted regression instead of their proposed supervised learning approach perform on top of data collected from a policy iteration based exploration policy? But the results seem quite promising.

I would also appreciate some more clarity in terms of writing and presentation.  While this is done in the appendix, I would suggest making references with regards to the supervised learning section (Sec 5) more explicit and refining some of the text in the paper, although this is a minor point.

**Experience Assessment:**

I have published one or two papers in this area.

**Review Assessment: Checking Correctness Of Derivations And Theory:**

I assessed the sensibility of the derivations and theory.

**Review Assessment: Checking Correctness Of Experiments:**

I assessed the sensibility of the experiments.

**Review Assessment: Thoroughness In Paper Reading:**

I read the paper at least twice and used my best judgement in assessing the paper.

---

> ### Author Response · Authors · 2019-11-11
> **Response to Reviewer #3**
>
> We thank the reviewer for the review and comments for improving the paper. We conducted more experiments including using SIL supervised loss in our off-policy framework, two more off-policy policy gradients baselines (SIL in all games and ACER in ablation study) to address your concerns. Please find the response as follows:
>
> Q1: The Assumption of the existence of UNOP seems strong.
> The reviewer is correct that assumption 2 regarding the existence UNOP may not be satisfied in practice, while empirically, this does not affect our performance. Theoretically, this assumption leads to a stationary distribution for stable optimization and optimality preserving. Furthermore, in sec 9.9, we show that assumption 2 can be satisfied in a range of tasks.
>
> Q2: The novelty of the reduction from RL to SL? how would the other methods such as self-imitation or reward weighted regression instead of their proposed supervised learning approach perform on top of data collected from a policy iteration based exploration policy?
>
> The key contribution of the proposed reduction from RL to SL is that it SIMULTANEOUSLY achieves optimality preserving (unbiasedness), variance reduction, stationary target (UNOP) and stable optimization for off-policy learning and we don’t require the access of Oracle (expert policy, human demonstration). Even though prior arts address some of the properties, but we noted none provides an integrated solution as we do in this paper.  We also summarized this in the related work, last paragraph and in Sec 9.1.
>
> More specifically, comparing to self-imitation learning (SIL), the advantage of the proposed method is that we preserve the optimality of the original RL objective. The SIL optimizing the lower bound of soft Q-learning may not guarantee the optimality. Comparing to reward weighted regression, the proposed method can learn in an off-policy way, while reward weighted regression has an on-policy requirement in the expectation step.
> The proposed framework is also flexible in terms of incorporating other exploration methods. In SIL, although SIL objective is off-policy, the value functions and policy share the same set of policy/value function parameters as A2C, which is on-policy.
>
> Regarding incorporating SIL and RWR into our framework, since their supervised loss also coupled with an on-policy counterpart, it is theoretically not clear what will it lead to if we drop on-policy part. To verify this, we evaluate IQN explore + SIL loss (eq1-3 in the SIL paper), the performance stuck at -20 for 2M steps on Pong, across five random seeds.
>
> Regarding the comparison of different supervised approaches in the same proposed off-policy framework, we compared hinge loss (RPG eq (6)) and cross-entropy loss (eq 69) and show RPG lead to better sample-efficiency.
> Regarding comparison with SIL, we add a comparison with SIL in all games (Figure 2). RPG is shown to be a more effective approach.
>
> Q3: The paper currently lacks intuition about the effectiveness of their policy gradient approach on top of the data collected from a DQN-based agent.
> As discussed in the paper (Fig 1), the intuition of the effectiveness of the proposed approach is the separation of the exploration stage and exploitation stage. The exploration stage should use the most effective method for collecting diversified near-optimal trajectories. In the Atari games, we found DQN-based agent is more effective in exploring good trajectories. The proposed method is not restricted to use DQN-based agents for collecting data. For example, in Pong, we use policy gradient as an exploration agent.
>
> Q4: Since reward shaping is done at the trajectory level, why would we expect the supervised regression step to do better than the best trajectory in the data?
> We do not expect the learned policy would perform better than the best trajectory in the data. The main improvement of sample-efficiency comes from the more effective exploitation, as we discussed in Section 6. The off-policy learning is to imitate the good trajectories and generalize to similar ones as well. In this case, the generalization is defined the same as in supervised learning.
>
> Q5: Can the same Q-learning based method perform better if one controls for the number of gradient updates?
> This question is closely related to whether we use a higher replay ratio to improve the sample efficiency, which may cause an unfair comparison.  The number of gradient updates of baselines was tuned by the dopamine framework. For our method, we keep the same amount of updates as the baseline per interaction. Therefore, the improvement of sample-efficiency doesn’t come from an increasing number of updates. Please also refer to section 9.12 for the description of hyperparameters.

---

### Official Review · AnonReviewer1 · 2019-10-23
**Official Blind Review #1**

**Rating:** 6

**Review:**

This work first establishes the connection of maximizing the lower bound of accumulated reward and supervised learning on near-optimal policies. Then it proposes a general framework for policy learning: during the exploration stage, the agent will collect near-optimal trajectories while in the exploitation stage, the agent will perform supervised learning on the collected data. Under this framework, the author argues that the ranking loss could outperform the state-of-the-art on the Atari benchmark.

The overall idea is intuitive yet interesting, and the empirical result is quite impressive.

Some questions which I think the paper could discuss more:
- In the paper, the near-optimal policy is defined with an absolute threshold, which could be task/environment-specific. I am wondering whether the author tried to set the `near-optimal policy` as a relative value (in the current replay buffer). Then the hyper-parameter could be shared.
- I think some empirical comparisons of the gradient variance (i.e., for Corollary 2) will be more demonstrative, although I could imagine that the near-optimal trajectories will have smaller variance.
- The choice of C could be tricky in the method as the whole algorithm is highly depending on it. How does the author choose C? If C is tuned for each environment, I am not sure whether it is a fair comparison with C51/Rainbow/IQN.
- More algorithm training details on the experimental setting (like the hyper-parameters) are needed.

======
From eq (7) -> eq (8): Does the Taylor expansion near $\lambda_{ji}=0$ make sense?


**Experience Assessment:**

I have read many papers in this area.

**Review Assessment: Checking Correctness Of Derivations And Theory:**

I assessed the sensibility of the derivations and theory.

**Review Assessment: Checking Correctness Of Experiments:**

I carefully checked the experiments.

**Review Assessment: Thoroughness In Paper Reading:**

I read the paper at least twice and used my best judgement in assessing the paper.

---

> ### Author Response · Authors · 2019-11-11
> **Response to Reviewer #1**
>
> We thank the reviewer for his/her review and comments for improving the paper.
>
> Q1: The choice of trajectory reward threshold C:
> C is a task-specific parameter that we choose according to the best performance of state-of-the-art. We strongly agree it would be a meaningful future work to study how to drop C (or using some dynamic strategy) while still maintain a stationary target for off-policy learning that preserves optimality. However, the usage of C benefits the method in many ways: stable training, unbiasedness, variance reduction. Additional tuning parameters like C has appeared in many learning approaches, e.g., In entropy regularized RL, the methods can be sensitive to the reward-scale; we also need to tune the extra temperature parameter. In ridge regression, we need to tune the extra l2 regularization parameter comparing to the linear regression. We thus believe that, even with C as an additional tuning parameter, it is fair to compare with baselines.
>
> Q2: More training details on the experimental setting:
> We keep the same hyperparameter setting, network structure as baselines such as batch size 32, update frequency 4, etc. We add sec 9.12 to describe the hyperparameters and more details.
>
> Q3: Regarding gradient variance.
> Many existing approaches either shape the reward scale (clipping or normalization) and/or clip the gradient norm. The investigation of the gradient variance of the original algorithm (without these empirical tricks) does not reflect the actual variance they encountered. The gradient variance of their actual implemented algorithms will be much smaller than the original algorithm.
> In RPG, the trajectory reward is shaped as described in Def 3 and no gradient clipping or other extra trick is needed. We agree that the investigation of gradient variance is important and it would be an interesting future work while it is not the focus of this work.
>
> Q4: From Eq 7 to Eq 8.
> We use Taylor expansion to approximate the log(1+e^x) at x = 0. This approximation would be more accurate if |x| is small. Since \lambda_ij denotes the relative order of action i and action j, its absolute value is not important. Therefore we can restrict the range of \lambda_ij to be small.

---

### Author Response · Authors · 2019-11-15
**Summary of updates**

We would like to thank all the reviewers for their helpful comments. We have provided responses to all of the reviewers, and have updated our paper accordingly. In summary, we would like to highlight the following changes:
1) Experiments: we added two additional baselines (SIL for all games and ACER in ablation study) for comparison, this further verifies the effectiveness of the proposed off-policy approach comparing to the state-of-the-art.
2) Writing: we improved the presentation of the materials by adding Sec 9.12 to show more training details, adding  Def 1, remark 1 to clarify the relative action values. We improved the writing of assumption 1, refine the proof in Sec 9.7,  adding more explanation on the limitation of the proposed method in Sec 9.11 and related works mentioned by reviewers.

---

### Public Comment · ~Assaf_Hallak1 · 2021-11-10
**Assumption 1**

Assumption 1 seems to be problematic as stated - obviously, $e_{ij}$ and $e_{ji}$ are dependent as both cannot happen simultaneously. Even if you redefine $E$ to have a specific $i$ index (i.e. $E_i$ = {e_{ij}: j \neq i} the assumption is very unlikely to be true - think for example of a uniform distribution over $N$ items, if we know action $a_i$ is higher ranked than $a_j$ then $a_i$ cannot be the lowest ranked one which increases the probability $a_i$ is higher than any other $a_k$ (as $a_k$ can be the lowest ranked one).

---

> ### Author Response · Authors · 2021-11-12
> **The independence is conditioned on MDP and policy.**
>
> Definition of conditional independence of a and b: P(a|b,c) = P(a|c), in our case: p(e_{12}|e_{21},M, \pi) = p(e_{12}|M, \pi).
> Please also refer to Appendix 10.10.

---

> > ### Public Comment · ~Assaf_Hallak1 · 2021-11-17
> > **Shouldn't**
> >
> > $p(e_{12}|e_{21},M, \pi) = 0 \neq p(e_{12}|M, \pi)$
> >
> > since e_{12} and e_{21} cannot happen simultaneously, but e_{12} can generally happen.

---

> > > ### Author Response · Authors · 2021-11-18
> > > **reply**
> > >
> > > assume deterministic policy \pi,  a_2 = argmax_j \lamda(a_j|s),  p(e_{12}|e_{21},M, \pi) = p(e_{12}|M, \pi) = 0.

---

### Decision · Program_Chairs · 2019-12-19

**Decision:**

Accept (Poster)

**Comment:**

The paper introduces a novel and effective approach to policy optimization.  The overall contribution is sufficient to merit acceptance.  Nevertheless, the authors should improve the presentation and experimental evaluation in line with the reviewer criticisms.  The criticisms of AnonReviewer2 in particular should not be neglected.  Regarding the theory, I agree with AnonReviewer3 that the UNOP assumption is too limiting.  The paper would be much stronger if this assumption could be significantly weakened, or better justified.